# T2SMark: Balancing Robustness and Diversity in Noise-as-Watermark for Diffusion Models

**Jindong Yang**[1,2], **Han Fang**[3*], **Weiming Zhang**[1,2], **Nenghai Yu**[1,2], **Kejiang Chen**[1,2*]
[1]University of Science and Technology of China
[2]Anhui Province Key Laboratory of Digital Security
[3]National University of Singapore
dx929@mail.ustc.edu.cn, fanghan@nus.edu.sg
{ynh, zhangwm, chenkj}@ustc.edu.cn

## Abstract

Diffusion models have advanced rapidly in recent years, producing high-fidelity images while raising concerns about intellectual property protection and the misuse of generative AI. Image watermarking for diffusion models, particularly Noise-as-Watermark (NaW) methods, encode watermark as specific standard Gaussian noise vector for image generation, embedding the infomation seamlessly while maintaining image quality. For detection, the generation process is inverted to recover the initial noise vector containing the watermark before extraction. However, existing NaW methods struggle to balance watermark robustness with generation diversity. Some methods achieve strong robustness by heavily constraining initial noise sampling, which degrades user experience, while others preserve diversity but prove too fragile for real-world deployment. To address this issue, we propose T2SMark, a two-stage watermarking scheme based on Tail-Truncated Sampling (TTS). Unlike prior methods that simply map bits to positive or negative values, TTS enhances robustness by embedding bits exclusively in the reliable tail regions while randomly sampling the central zone to preserve the latent distribution. Our two-stage framework then ensures sampling diversity by integrating a randomly generated session key into both encryption pipelines. We evaluate T2SMark on diffusion models with both U-Net and DiT backbones. Extensive experiments show that it achieves an optimal balance between robustness and diversity.

## 1 Introduction

In recent years, diffusion models [1–4] have achieved groundbreaking advances in generative AI, especially in image synthesis. Owing to their iterative denoising process, these models can produce highly realistic images. As their architecture scales and training datasets expand, diffusion models continually improve in both generation quality and controllability, driving rapid progress in AIGC technologies.

However, this swift evolution also brings new challenges: the authenticity of generated content is increasingly difficult to verify, and diffusion models have become potent instruments for spreading disinformation and manipulating public opinion. For example, during the 2024 U.S. presidential election, U.S. officials accused Russia and other countries of using generative AI to fabricate fake news,

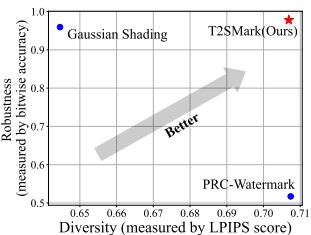

Figure 1: T2SMark strikes an optimal balance between robustness and diversity.

---

*Corresponding authors.

39th Conference on Neural Information Processing Systems (NeurIPS 2025).

images, and videos to influence voter sentiment and disrupt the electoral process[2]. Meanwhile, service providers invest vast amounts of data and computing power to train these models and urgently need effective mechanisms to safeguard their intellectual property against unauthorized use. Together, these concerns have increased the demand for robust solutions to protect the copyright and trace the provenance of AI-generated images, making this a focal point of current academic research.

To address these challenges, watermarking presents a fundamental solution and has shown considerable promise. Among various techniques, Noise-as-Watermark (NaW) methods, including Gaussian Shading (GS) [5] and PRC-Watermark (PRCW) [6], stand out. At its core, NaW maps watermark codewords onto a standard Gaussian noise vector, uses it as the initial noise for image generation, and thus embeds watermark information while preserving the noise distribution. For extraction, one simply inverts the diffusion process to recover the embedded noise and subsequently decodes it to retrieve the watermark. The reversible bijection offered by the forward and backward diffusion processes between high-entropy Gaussian vectors and the low-entropy image manifold allows NaW methods to achieve high-capacity, lossless embedding for individual images without the need for supplementary training.

The embedding process in most Noise-as-Watermark (NaW) methods comprises two core steps: discrete encoding and continuous sampling. The former step converts the watermark message into codewords, while the latter transforms those discrete binary codes into continuous noise. The challenge in discrete encoding lies in balancing the need for randomness, which makes the watermark difficult to predict, with the need for robustness against various distortions — goals that are often contradictory. Continuous sampling, on the other hand, is critical for maintaining imperceptibility by precisely preserving the initial Gaussian noise distribution. Current methods make different trade-offs: GS [5] employs a simple repetition code for high robustness but lacks randomness, depending entirely on the sampling process for variation. PRCW [6], while using a pseudorandom error-correcting code [7] for better undetectability, exhibits weak robustness and can fail when subjected to common inversion errors.

To overcome these limitations and achieve a balance of both robustness and diversity, we propose **T2SMark**, *a novel two-stage watermarking scheme built upon Tail-Truncated Sampling (TTS)*. While previous efforts have concentrated primarily on discrete encoding strategies, we observed that Gaussian samples closer to the origin are significantly more vulnerable to sign errors when noise is present. This finding inspired our TTS approach: instead of a simple positive/negative mapping for bits, TTS divides the Gaussian distribution into three distinct areas: one for bit-0, one for bit-1, and an undecided region (see Figure 2). Information is embedded only within the more reliable tail regions, with the central region being sampled randomly to compensate for diversity and maintain the overall distribution. Additionally, our two-stage framework introduces controlled randomness via hierarchical key encryption. A static master key encrypts a randomly generated session key in the first stage, and this session key is then used to encrypt the actual watermark bits in the second stage. This layered encryption randomizes both parts of the watermark codewords, thereby further ensuring the generation diversity. Finally, we improve detection and decoding by employing multidimensional projections of the reconstructed Gaussian noise, allowing us to fully leverage its rich continuous information.

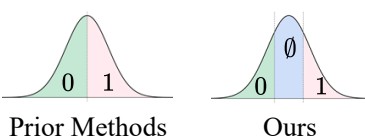

Figure 2: Tail-Truncated Sampling divides the distribution into three regions: bit-0, bit-1 and an undecided zone ($\emptyset$).

Evaluated on diffusion models [3, 8] with both U-Net [9] and DiT [4] backbones, T2SMark demonstrates an optimal trade-off between watermark robustness and generative diversity.

## 2 Related Work

### 2.1 Diffusion Models

Diffusion models [1–4] iteratively corrupt data by injecting Gaussian noise and train a denoising network to reverse this process. The canonical DDPM framework [1] employs a stochastic Markov chain for multi-step sampling, delivering outstanding sample quality at the cost of slow inference.

---

[2]U.S. officials say Russia is embracing AI for its election influence efforts. NPR, September 23, 2024.

Building on DDPM, DDIM [2] reformulates the random process as a deterministic ODE, transforming sampling into an invertible mapping that both accelerates generation and enables inversion—making it the backbone of many inversion-based watermarking techniques [5, 6, 10]. Moreover, latent diffusion models (LDMs) [3] first compress high-dimensional data into a lower-dimensional latent space and then perform noise addition and denoising in that compact space. This strategy dramatically reduces computation while preserving generation quality and significantly boosting efficiency.

Inspired by the Transformer's success in NLP [11–13], researchers have introduced self-attention into diffusion models. Diffusion transformers (DiTs) [4] leverage the scalability of transformer architectures for image generation. Today, many leading diffusion pipelines [8, 14, 15] adopt DiT backbones, driving advances in high-fidelity and efficient synthesis.

Generation diversity is an essential metric for assessing a model's creative capacity. ODE-based samplers [2, 16, 17] achieve faster inference by discarding stochastic noise during denoising, which means that diversity depends entirely on the initial noise vector. As a result, any watermarking scheme that alters or constrains the initial noise sampling can have a direct effect on generation diversity.

## 2.2 Image Watermarking for Diffusion Models

Image watermarking methods for diffusion models can be divided into three main categories. 1) **Post-processing schemes** [18–21] embed a watermark directly into the generated image—using either traditional transform-domain techniques [18] or deep learning–based methods [19–21]—without modifying the sampling process. While this preserves model diversity, it inevitably degrades visual quality and provides only limited robustness. 2) **Fine-tuning schemes** [22, 23] inject watermarks by adapting either the diffusion model's denoiser or, in a latent diffusion setup, the VAE [24] backbone; these methods maintain sample fidelity but require costly weight updates as model architectures grow. 3) **Inversion-based schemes** [5, 6, 10] embed information into the initial Gaussian noise sample and recover it by inverting the diffusion process. This category includes perturbation-based approaches such as Tree-Ring [10] and RingID [25], which adds robust Fourier-domain patterns at the expense of distributional bias, and Noise-as-Watermark (NaW) techniques such as Gaussian Shading (GS) [5] and PRC-Watermark (PRCW) [6], which use distribution-preserving sampling to achieve high embedding capacity without shifting the noise distribution. Our T2SMark method also falls into this NaW paradigm. Building upon the principles of Tree-Ring, other methods such as ROBIN [26] and ZoDiac [27] focus on ownership detection rather than message embedding. These are 0-bit watermarking schemes, designed solely to verify if an image was generated by a specific model. They leverage gradient-based optimization to embed an imperceptible signature, achieving impressive fidelity and competitive robustness. However, their design philosophy differs significantly from our approach. By not needing to be strictly identical to an unwatermarked original, training-free NaW methods can fully leverage the initial noise space to achieve a high capacity (e.g., 256-bit) for traceability. Given these fundamental differences in goals (detection vs. traceability) and methodology (optimization-based vs. training-free), we do not conduct a direct comparison with them in this paper.

## 3 Method

### 3.1 Threat Model

**Scenarios.** This study considers four key parties: the provider, who operates a closed API image generation service maintaining proprietary model weights and training data; the compliant user, who adheres to platform policies and utilizes the outputs legitimately; the unauthorized user, who illicitly obtains images to claim them as their own; and the malicious user, who exploits the API to generate and disseminate harmful or deceptive content.

**Detection.** The provider embeds a single-bit watermark into every image generated and serves through the API. Both compliant and malicious users receive only watermarked outputs, whereas unauthorized users have no means to obtain unmarked images. Even after typical manipulations, such as compression or cropping, our extraction algorithm robustly recovers the embedded watermark bit, thereby furnishing incontrovertible proof of the provider's legitimate copyright while simultaneously signaling the image's synthetic origin.

**Traceability.** Each API account is uniquely assigned an identity watermark. In instances of misuse, extracting the embedded identity watermark from a suspect image (potentially after data augmentation designed to evade tracing) enables the provider to match it against the account database, thus pinpointing the source, deterring unauthorized distribution, and holding malicious actors accountable.

## 3.2 Overview

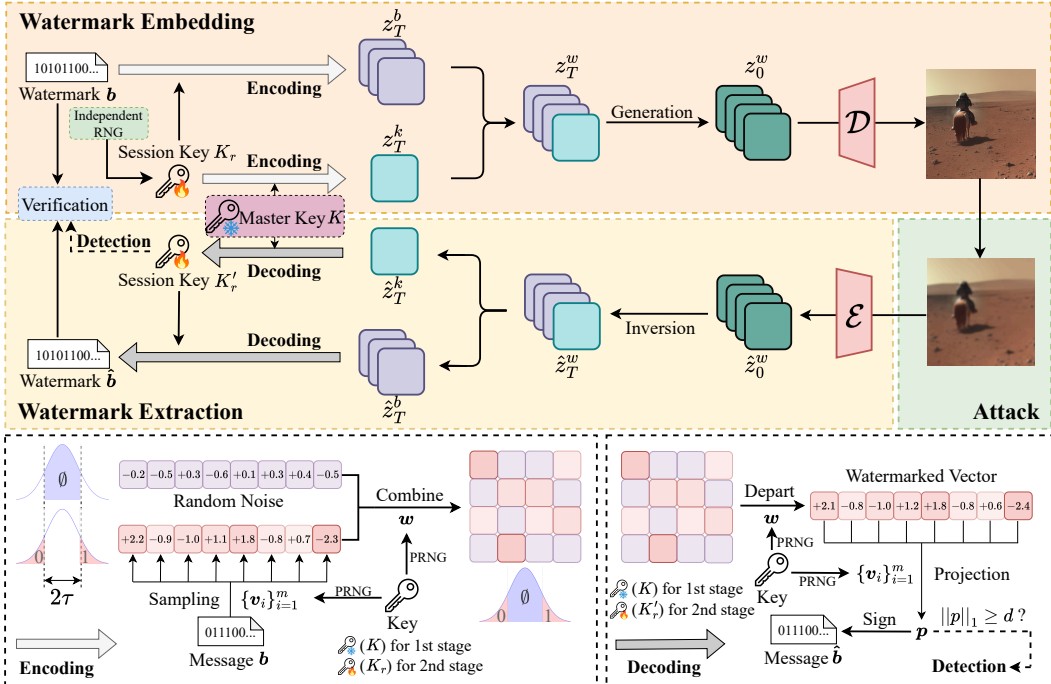

Figure 3: The framework of T2SMark. Building on the NaW paradigm, Tail-Truncated Sampling enhances robustness while the two-stage structure introduces controlled randomness.

T2SMark can be clearly illustrated by Fig. 3. We first split the noise vector into two segments: the first encodes a random session key under a fixed master key, and the second uses that session key to encode the actual watermark bits. Because both segments depend on the random key, the full vector remains randomized.

Formally, in the single-stage setting, we treat the $n$-dimensional noise space as a direct sum of orthogonal subspaces. The key pseudorandomly selects a hyperplane through the origin that splits each subspace into two half spaces for binary encoding. During sampling, we employ *Tail-Truncated Sampling*: points are drawn along each hyperplane's normal and kept at least a distance $\tau$ from the boundary. This thresholding carves out a robust watermark-encoding subspace of larger-magnitude vectors that resist cross-boundary perturbations, while the remaining dimensions carrying no bits retain randomness.

At extraction, we reconstruct the noise via an inversion algorithm [2] and project it onto the same set of hyperplane normals. Each bit is recovered from the sign of its projection, which identifies the half-space containing the sample, while detection uses the projection's $L_1$ norm as a confidence measure. This approach fully exploits the continuous structure of the noise vector. The detailed scheme follows in subsequent sections.

## 3.3 Watermark Encoding

Here, *encoding* means mapping the watermark into a continuous noise vector, not just discrete coding. Let $n \in \mathbb{N}$ be the initial noise dimension, $\tau \geq 0$ the truncation threshold for Tail-Truncated Sampling, $m \in \mathbb{N}$ the watermark length, and $\boldsymbol{b} \in \{\pm 1\}^m$ the bit-vector.

Based on the threshold $\tau$, we determine the expected number of tail-sampled dimensions $k$ and the per-bit subspace dimension $r$:

$$k = 2\,\Phi(-\tau)\,n, \qquad r = \lfloor k/m \rfloor, \tag{1}$$

Using a secret key $K$ as the seed for a PRNG (Pseudorandom Number Generator), we generate a set of $m$ pseudorandom vectors $\{\boldsymbol{v}_j\}_{j=1}^{m}$, where each $\boldsymbol{v}_j \in \{-1, 0, +1\}^n$. These vectors are designed with orthogonal, non-overlapping supports and serve as the normal vectors for the bit-encoding hyperplanes within the watermark embedding subspace:

$$\|\boldsymbol{v}_j\|_0 = r, \quad \operatorname{supp}(\boldsymbol{v}_j) \cap \operatorname{supp}(\boldsymbol{v}_{j'}) = \emptyset \quad (j \neq j'). \tag{2}$$

Next, we form the binary mask $\boldsymbol{w} = \sum_{j=1}^{m} |\boldsymbol{v}_j| \in \{0,1\}^n$. This mask partitions the $n$ dimensions into the watermark-encoding subspace ($w_i = 1$) and the random-noise subspace ($w_i = 0$). Tail-Truncated Sampling (TTS) is applied to sample components for the noise vector $\boldsymbol{z}$:

$$z_i \sim \begin{cases} \mathcal{TN}(0, 1; [-\tau, \tau]), & w_i = 0, \\ \mathcal{TN}(0, 1; (-\infty, -\tau] \cup [\tau, \infty)), & w_i = 1. \end{cases} \tag{3}$$

where $\mathcal{TN}(\mu, \sigma^2; I)$ denotes the normalized truncation of $\mathcal{N}(\mu, \sigma^2)$ over interval $I$. The final watermarked noise vector $\boldsymbol{z}^w$ is then constructed by combining the signs corresponding to the bits in the tail-sampled dimensions with their magnitudes from $\boldsymbol{z}$, while incorporating the random samples from the central dimensions. This is achieved by:

$$\boldsymbol{z}^w = \boldsymbol{w} \odot |\boldsymbol{z}| \odot \sum_{j=1}^{m} b_j\,\boldsymbol{v}_j + (\boldsymbol{1}^n - \boldsymbol{w}) \odot \boldsymbol{z}, \tag{4}$$

where $\odot$ is the Hadamard product. The term $\sum_{j=1}^{m} b_j\,\boldsymbol{v}_j$ represents the directional encoding of the bits $\boldsymbol{b}$ within the subspace defined by $\{\boldsymbol{v}_j\}_{j=1}^{m}$.

### 3.4 Watermark Decoding

Upon reconstructing the watermarked noise $\widehat{\boldsymbol{z}}^w$ via diffusion inversion [2], we regenerate the set of vectors $\{\boldsymbol{v}_j\}_{j=1}^{m}$ using the same secret key $K$. The projection vector $\boldsymbol{p} \in \mathbb{R}^m$ is computed by taking the dot product of $\widehat{\boldsymbol{z}}^w$ with each vector $\boldsymbol{v}_j$:

$$p_j = \langle \widehat{\boldsymbol{z}}^w, \boldsymbol{v}_j \rangle, \quad \text{for } j = 1, \ldots, m. \tag{5}$$

The watermark bit-vector $\widehat{\boldsymbol{b}}$ is recovered directly from the signs of the projection vector components:

$$\widehat{\boldsymbol{b}} = \operatorname{sign}(\boldsymbol{p}), \tag{6}$$

where $\operatorname{sign}(\cdot)$ is applied elementwise to the $m$-dimensional vector $\boldsymbol{p}$. With a per-bit encoding dimension $r$ less than $\lfloor n/m \rfloor$ (indicating reduced repetition), a higher signal-to-noise ratio (SNR) is achieved by TTS, theoretically leading to an even lower probability of bit error under the AWGN assumption (see Appendix A).

### 3.5 Two-Stage Watermark

We split the $n$-dimensional noise into two segments of dimensions $n_k$ and $n_b$ (with $n_k + n_b = n$), often corresponding to channelwise partitioning. First, using the master key $K$, we sample an $n_k$-dimensional vector $\boldsymbol{z}^k$ that encodes a random session key $K_r$. Next, with $K_r$, we sample an $n_b$-dimensional vector $\boldsymbol{z}^b$ containing the watermark bits $\boldsymbol{b}$. The combined watermarked noise is given by $\boldsymbol{z}^w = \boldsymbol{z}^k \| \boldsymbol{z}^b$, where $(\cdot \| \cdot)$ denotes vector concatenation.

During extraction, we partition the reconstructed noise $\widehat{\boldsymbol{z}}^w$ into $\widehat{\boldsymbol{z}}^k$ and $\widehat{\boldsymbol{z}}^b$. We first recover $K_r'$ from $\widehat{\boldsymbol{z}}^k$ via the master key $K$, and then retrieve the watermark $\widehat{\boldsymbol{b}}$ from $\widehat{\boldsymbol{z}}^b$ using $K_r'$. The session key $K_r$ thus serves a dual role: embedded payload in the first segment and key for the second, injecting randomness across $\boldsymbol{z}^w$ and preserving generation diversity.

### 3.6 Detection and Traceability

Detection relies primarily on the first stage due to the risk of error propagation in two-stage decoding. Leveraging the larger norms of projections afforded by TTS, the test statistic is computed based on the $L_1$ norm of the first-stage projection vector:

$$l = \left\| \boldsymbol{p}_k \right\|_1, \quad (\boldsymbol{p}_k)_j = \langle \widehat{\boldsymbol{z}}^k, \boldsymbol{v}_{kj} \rangle, \quad j = 1, \dots, m_k. \tag{7}$$

In this equation, $m_k$ denotes the size of the session key. $\boldsymbol{p}_k$ is the projection vector obtained by projecting $\widehat{\boldsymbol{z}}^k$ onto the first stage's normal vectors $\{\boldsymbol{v}_{kj}\}_{j=1}^{m_k}$. This statistic $l$ is compared against a threshold $d$ calibrated to a target false-positive rate.

For traceability, full two-stage decoding is performed to recover the complete embedded bit-vector $\widehat{\boldsymbol{b}}$. This decoded watermark is compared against the registered identity watermark database. The account whose assigned watermark yields the highest similarity score (e.g., match count or Hamming distance) is identified as the image's originator, enabling misuse tracing and accountability.

## 4 Experiments

### 4.1 Experimental Setup

**Implementation Details.** Our image generation backbone is Stable Diffusion v2.1 (SD v2.1) [3], configured with a guidance scale of 7.5, 50 DDIM denoising steps, and a fixed $512 \times 512$ output resolution. T2SMark employs a 16-bit session key and a 256-bit watermark. We set the truncation threshold to $\tau = 0.674$. The session key is embedded in the first channel of the initial noise. This is determined empirically in our parameter selection study (see Appendix B.2). All the experiments are implemented in PyTorch 2.4.1 and run on a single NVIDIA RTX A6000 GPU.

**Baselines.** We compare against three categories of existing methods. Traditional post-processing transforms include dwtDct [18], dwtDctSvd [18], and the learning-based RivaGAN [19]; fine-tuning approaches are represented by Stable Signature [22]; Inversion-based Schemes include Tree–Ring (TRW) [10], Gaussian Shading (GS) [5] and PRC-Watermark (PRCW) [6]. For all inversion-based methods, we perform 10-step DDIM inversion [2]. During inversion, we employ an empty prompt and fix the guidance scale at 1 to simulate unknown prompt conditions. To ensure fair capacity, dwtDct, dwtDctSvd, Gaussian Shading, and PRC-Watermark all embed 256 bits. RivaGAN and Stable Signature use 32 bits and 48 bits respectively, following their official implementations.

**Evaluation.** We evaluate on MS-COCO-2017 [28] dataset (COCO) and Stable-Diffusion-Prompt[3] dataset (SDP). For robustness, we compare the TPR at a fixed FPR $= 10^{-6}$ in the detection setting and per-bit accuracy in the traceability setting. For each method, we sample 500 prompts from the SDP training split, generate 500 watermarked images, apply nine different distortions (see Figure 4), and then perform detection and traceability.

We assess generation diversity via LPIPS [29]. For each non–post-processing method, we generate 10 images for each of 1,000 COCO test prompts with a fixed master key and watermark, compute LPIPS over the 45 unique pairs per prompt, and report the overall mean. Post-processing methods leave the core generation unchanged and are therefore omitted from this evaluation.

For visual quality, we report the CLIP score [30] and FID [31]. We run 10 independent trials: in each trial, we fix a single key–watermark pair (simulating one user), and generate 1 000 images from COCO's test prompts for the CLIP score and FID. We then perform two-sample $t$-tests to assess the impact of the watermark on quality. The hypotheses are $H_0 : \mu_s = \mu_0, \quad H_1 : \mu_s \neq \mu_0$, where $\mu_s$ and $\mu_0$ denote the mean FID or CLIP score computed over multiple sets of watermarked and clean images, respectively. A lower $t$-value indicates stronger support for $H_0$. Details about the $t$-test can be found in Appendix B.3.

### 4.2 Main Results

Results for the different methods are presented in Table 1. T2SMark demonstrates superior performance in the traceability scenario while achieving detection performance comparable to the best

---

[3]Stable-Diffusion-Prompts

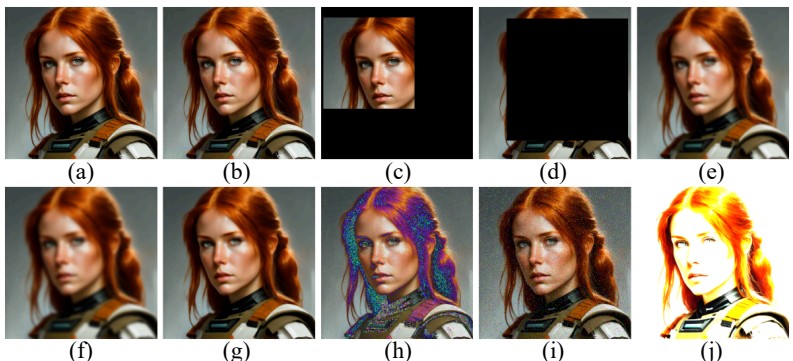

Figure 4: The watermarked image is attacked by different types of noise. (a) Watermarked image. (b) JPEG, $QF = 25$. (c) 60% area Random Crop (RandCr). (d) 80% area Random Drop (RandDr). (e) 25% Resize and restore (Resize). (f) Gaussian Blur, $r = 4$ (GauBlur). (g) Median Blur, $k = 7$ (MedBlur). (h) Gaussian Noise, $\mu = 0$, $\sigma = 0.05$ (GauNoise). (i) Salt and Pepper Noise, $p = 0.05$ (S&PNoise). (j) Brightness, $factor = 6$.

Table 1: Evaluation results of watermarking methods, including detection (TPR), traceability (Bit Acc.), generation diversity, and visual quality (CLIP score and FID). TPR and Bit Acc. are shown as Clean/Adv. Standard errors and $t$-values are shown for the CLIP score and FID.

| Method | TPR | Bit Acc. | Diversity ↑ | CLIP Score ($t \downarrow$) | FID ($t \downarrow$) |
|---|---|---|---|---|---|
| SD v2.1 [3] | – | – | 0.7072 | 0.3224±0.0010 (–) | 56.8132±0.48 (–) |
| dwtDct [18] | 0.922/0.173 | 0.8177/0.5744 | – | 0.3206±0.0010 (3.7233) | 56.3471±0.48 (2.0270) |
| dwtDctSvd [18] | **1.000**/0.471 | 0.9988/0.6800 | – | 0.3209±0.0010 (3.2150) | 56.0986±0.48 (3.1220) |
| RivaGAN [19] | 0.914/0.436 | 0.9823/0.8666 | – | 0.3209±0.0010 (3.2150) | 56.1893±0.48 (2.7332) |
| StableSig [22] | **1.000**/0.418 | 0.9981/0.7462 | 0.6917 | 0.3235±0.0007 (2.6063) | 56.0423±0.38 (3.7020) |
| TRW [10] | **1.000**/0.907 | –/– | 0.6943 | 0.3210±0.0007 (3.6365) | 58.2667±0.36 (6.9293) |
| GS [5] | **1.000**/0.998 | **1.0000**/0.9548 | 0.6446 | 0.3242±0.0027 (1.7557) | 58.1377±1.19 (3.0807) |
| PRCW [6] | **1.000**/0.294 | 0.6494/0.5024 | **0.7074** | 0.3218±0.0009 (1.4369) | 56.8975±0.38 (**0.4056**) |
| T2SMark | **1.000**/0.998 | **1.0000**/0.9754 | 0.7069 | 0.3227±0.0008 (**0.5081**) | 56.9317±0.42 (0.5490) |

method, GS [5]. In contrast, PRCW [6] performs worst in traceability and registers a detection TPR below 30% under adversarial conditions, which is too fragile for real-world deployment. More detailed robustness results under various noise types can be found in Appendix B.4.

With respect to generation diversity, measured by LPIPS [29], PRCW achieves the highest score. T2SMark trails closely behind, with a difference of less than $10^{-3}$, representing a negligible gap. GS exhibits the lowest diversity score, which is attributable to its use of a fixed codeword for each user, while Stable Signature and TRW also experience noticeable reductions in diversity.

Regarding image quality, only T2SMark and PRCW consistently satisfy the no-degradation criterion across both evaluation tests. Although GS achieves a competitive CLIP score [30], its FID [31] deviates significantly from the no-watermark baseline. Furthermore, GS exhibits a noticeably larger standard deviation in the CLIP score and FID, suggesting that its generation quality is sensitive to the specific watermark and key used. This sensitivity poses a challenge for the user experience, potentially leading to inconsistent generation quality when different user accounts (with different watermarks/keys) are involved. Considering all the evaluated metrics, T2SMark achieves the best overall balance.

## 4.3 Undetectability

To evaluate watermark undetectability, we trained a ResNet-18 classifier [32] to distinguish between watermarked and non-watermarked images. We assessed four inversion-based methods using a fixed key and watermark for each. The training set for each method consisted of 8,000 watermarked and 8,000 clean samples, while the test set contained 500 samples each. Training was conducted for 10 epochs with a batch size of 128 and a learning rate of $1 \times 10^{-4}$. Table 2 reports the test accuracy. The results indicate that TRW [10] and GS [5] are relatively easy to detect. Although

PRCW [6] demonstrates the highest level of undetectability, we observe that it is not entirely immune to detection. T2SMark achieves the second-best performance and can also be considered difficult to detect, demonstrating a high level of imperceptibility.

Table 2: Undetectability of different inversion-based watermarking methods, measured by detection accuracy (Det. Acc.). A lower accuracy indicates better undetectability.

|  | TRW [10] | GS [5] | PRCW [6] | T2SMark |
|---|---|---|---|---|
| Det. Acc. | 0.971 | 0.994 | **0.532** | 0.578 |

## 4.4 Generalizability

We evaluate T2SMark along with other inversion-based watermarking methods on the Stable Diffusion v3.5 Medium model (SD v3.5M) [8], which employs DiT [4] as its denoising network and features a 16-channel latent space. The detailed settings can be found in Appendix B.1.

The results in Table 3 show that SD v3.5M delivers superior generative quality so that all methods maintain strong visual fidelity. TRW's robustness declines markedly compared with its performance on SD v2.1, and GS experiences a similar loss of diversity. PRCW improves traceability but still trails the best method by a wide margin. In contrast, T2SMark achieves the best balance of robustness and diversity, and its undetectability is virtually indistinguishable from that of PRCW.

Table 3: Evaluation results of inversion-based watermarking methods on SD v3.5M, including detection (TPR), traceability (Bit Acc.), detection accuracy (Det. Acc.), generation diversity, and visual quality (CLIP score and FID). The TPR and Bit Acc. are shown as Clean/Adv. Standard errors and $t$-values are shown for the CLIP score and FID.

| Method | TPR | Bit Acc. | Det. Acc. ↓ | Diversity ↑ | CLIP Score ($t$ ↓) | FID ($t$ ↓) |
|---|---|---|---|---|---|---|
| SD v3.5M [8] | − / − | − / − | - | 0.6113 | 0.3498±.0010 (-) | 55.7627±.56 (-) |
| TRW [10] | 0.878 / 0.318 | − / − | 0.984 | 0.5924 | 0.3493±.0005 (1.3493) | 55.9084±.53 (0.5699) |
| GS [5] | **1.000** / **0.990** | 0.9994 / 0.9663 | 0.991 | 0.5176 | 0.3502±.0004 (0.9943) | 56.2020±.67 (1.5043) |
| PRCW [6] | 0.998 / 0.279 | 0.9920 / 0.6067 | **0.516** | 0.6096 | 0.3502±.0005 (0.9726) | 55.5673±.49 (0.7902) |
| T2SMark | **1.000** / 0.985 | **1.0000** / **0.9768** | 0.518 | **0.6102** | 0.3499±.0004 (**0.0991**) | 55.8121±.49 (**0.1983**) |

## 4.5 Ablation Study

In this section we conduct ablation experiments on SD v2.1 to evaluate robustness under varying hyperparameter settings. Unless otherwise specified, we generate 500 images on the SDP dataset and report the true positive rate at a fixed false positive rate of $10^{-6}$ alongside bit accuracy.

**Tail-Truncated Sampling.** We evaluate T2SMark both with and without Tail-Truncated Sampling (TTS), and additionally measure generation diversity over 1,000 prompts from the SDP dataset. The results in Table 4 demonstrate that TTS provides a substantial robustness boost while having only a negligible effect on diversity. This confirms TTS as a critical component of T2SMark.

Table 4: Performance of T2SMark both with and without Tail-Truncated Sampling.

|  | TPR (Clean/Adv.) | Bit Acc. (Clean/Adv.) | Diversity ↑ |
|---|---|---|---|
| w/o TTS | 1.000/0.996 | 0.9988/0.9307 | 0.6743 |
| w/ TTS | 1.000/0.998 | 1.0000/0.9754 | 0.6746 |

**Noise Intensities.** To further test the robustness, we conduct experiments using different intensities of noise. The results are shown in Figure 5(a-i). T2SMark is highly vulnerable to Gaussian noise—even at a low noise standard deviation of 0.1, its performance degrades dramatically, a weakness that has also been reported in experiments of other inversion-based methods [5, 10].

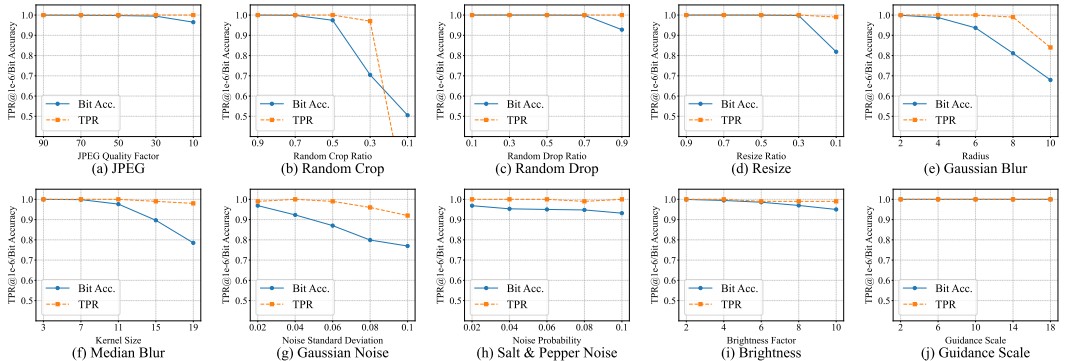

Figure 5: Ablation Studies.

**Guidance Scales.** Given diverse user preferences for prompt adherence, higher guidance scales enforce the original prompt more strictly, while lower scales allow greater creative freedom. Since Stable Diffusion v2.1 typically uses guidance scales between 5 and 15, our experiments span a wider range of 2-18. As shown in Figure 5(j), T2SMark's performance degrades only marginally under these settings.

**Inversion Steps.** We evaluate T2SMark with 5, 10, 25, 50, and 100 inversion steps (Table 5) and observe only minor performance fluctuations. Therefore, we can directly improves the extraction efficiency by reducing the number of inversion steps, as inversion is the primary bottleneck in NaW extraction pipelines.

Table 5: Impact of inversion steps on robustness performance. The TPR and Bit Acc. are shown as Clean/Adv.

| Inversion Steps | TPR | Bit Acc. |
|---|---|---|
| 5 | 1.000/0.996 | 1.0000/0.9708 |
| 10 | 1.000/0.997 | 1.0000/0.9764 |
| 25 | 1.000/0.996 | 1.0000/0.9764 |
| 50 | 1.000/0.998 | 1.0000/0.9745 |
| 100 | 1.000/0.997 | 1.0000/0.9730 |

Table 6: Performance under different capacity settings. The Bit Acc. is shown as Clean/Adv.

| Capacity (bits) | Bit Acc. |
|---|---|
| 256 | 1.0000/0.9754 |
| 384 | 1.0000/0.9595 |
| 512 | 1.0000/0.9437 |
| 768 | 0.9992/0.9145 |
| 1024 | 0.9968/0.8789 |

**Watermark Capacity.** We evaluate T2SMark using watermark capacity of 256, 384, 512, 768 and 1024 bits (Table 6). As expected, the robustness of T2SMark degrades with increased capacity. However, if the actual noise level in deployment is lower than that used in our experiments, a moderate increase in capacity remains tolerable.

Table 7: Performance under different session key sizes.

| | 8 | 16 | 24 | 32 |
|---|---|---|---|---|
| TPR (Clean/Adv.) | 1.000/0.998 | 1.000/0.998 | 1.000/0.996 | 1.000/0.991 |
| Bit Acc. (Clean/Adv.) | 1.0000/0.9776 | 1.0000/0.9754 | 1.0000/0.9687 | 1.0000/0.9481 |

**Session Key Size.** We evaluate T2SMark via random key sizes of 8, 16, 24, and 32 bits (Table 7). As the length of the random key increases, it becomes more difficult to extract the key perfectly in the first stage, which leads to cascading errors and a rapid drop in bit accuracy. While longer keys can theoretically enhance randomness, we find that a 16-bit key provides sufficient entropy for practical use—since it is unlikely that a single user would generate a number of images large enough to exhaust this space. Moreover, even with a 32-bit random key, T2SMark still maintains acceptable robustness.

## 5 Limitations

Despite its wonderful performance on diversity and robustness, T2SMark is subject to several limitations. A key set of challenges is shared among most NaW methods. First, the high robustness that allows for watermark recovery can be exploited in a forgery attack. An adversary could use a proxy model to invert the diffusion process, recover the watermarked noise vector, and then use this vector to generate new, forged images that appear authentic because they carry a valid watermark [33]. Second, T2SMark, like other NaW schemes, depends on an invertible, ODE-based sampling method (e.g., DDIM [2], DPM-Solver [16]). Without such a sampler, the initial noise cannot be reconstructed, preventing watermark recovery and limiting applicability to diffusion models that support this feature. Third, NaW methods are vulnerable to geometric distortions since their designs lack explicit mechanisms to resist it. Recent work like GaussMarker [34] offers effective solutions to this. Finally, there is a potential conflict with certain controllable generation methods that require modifying the core sampling logic, as this could interfere with the watermark embedding and detection process. In addition to these shared challenges, T2SMark has a unique vulnerability stemming from its key embedding strategy: Embedding the session key in the truncated tails concentrates energy in a way that introduces subtle distributional anomalies, which can be detected.

## 6 Conclusion

T2SMark presents a novel watermarking framework for diffusion-generated images. It leverages Tail-Truncated Sampling (TTS) to increase robustness and employs a two-stage key hierarchy to introduce controlled randomness and support generation diversity. Experiments on both the U-Net [9] and DiT [4] architectures demonstrate that T2SMark achieves superior traceability and competitive detection performance while effectively preserving diversity and image quality.

## Acknowledgments

This work was supported in part by the National Natural Science Foundation of China under Grant 62472398, Grant U2336206, Grant U2436601, and Grant 62402469.

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

## A    BEP Analysis under Tail-Truncated Sampling

We model decoding as a binary decision under additive white Gaussian noise (AWGN) with variance $\sigma^2$. Owing to the symmetry of the problem, we analyze the case corresponding to the transmission of bit 1 without loss of generality. Let $\mu(\tau)$ and $D(\tau)$ represent the mean and variance of a standard normal variable truncated in $[\tau, +\infty)$. Aggregating $r$ truncated samples yields a test statistic with an effective signal-to-noise ratio

$$\mathcal{S}(\tau) = \frac{\sqrt{r}\,\mu(\tau)}{\sqrt{D(\tau) + \sigma^2}}, \quad r = \lfloor 2\,\Phi(-\tau)\,n/m \rfloor \approx 2\,\Phi(-\tau)\,n/m. \tag{8}$$

In practice we approximate $\mathcal{S}(\tau)$ by

$$A(\tau) = a\,\frac{\sqrt{\Phi(-\tau)}\,\mu(\tau)}{\sqrt{D(\tau) + \sigma^2}}, \quad a = \sqrt{2n/m} \tag{9}$$

and the corresponding bit-error probability is

$$P_e(\tau) = \Phi\big(-A(\tau)\big), \tag{10}$$

where $\Phi(\cdot)$ is the standard normal CDF. For a tail-truncated Gaussian distribution,

$$\mu(\tau) = \mathbb{E}\,[Z \mid Z > \tau] = \frac{\phi(-\tau)}{\Phi(-\tau)}, \quad D(\tau) = \text{Var}\,[Z \mid Z > \tau] = 1 + \tau\,\mu(\tau) - \mu(\tau)^2, \tag{11}$$

where $\phi(\cdot)$ is the standard normal PDF.

To assess whether the truncation's SNR gain outweighs the redundancy loss for small $\tau > 0$, define

$$B(\tau) = \frac{\phi(-\tau)}{\sqrt{\Phi(-\tau)}}, \qquad C(\tau) = \frac{1}{\sqrt{D(\tau) + \sigma^2}}. \tag{12}$$

Then $A(\tau) = a\,B(\tau)\,C(\tau)$ and

$$P_e'(\tau) = -\phi\big(-A(\tau)\big)\,A'(\tau), \quad A'(\tau) = a\big[B'(\tau)\,C(\tau) + B(\tau)\,C'(\tau)\big].$$

First, at $\tau = 0$,

$$B(0) = \frac{\phi(0)}{\sqrt{\Phi(0)}} = \frac{1/\sqrt{2\pi}}{\sqrt{1/2}} = \frac{1}{\sqrt{\pi}} > 0, \tag{13}$$

$$C(0) = \frac{1}{\sqrt{D(0) + \sigma^2}} > 0, \tag{14}$$

since $D(0) \geq 0$ and $\sigma^2 > 0$. Next,

$$B'(0) = \frac{d}{d\tau}\big[\phi(-\tau)\,\Phi(-\tau)^{-1/2}\big]_{\tau=0} = \frac{1}{2}\,\frac{\phi(0)^2}{\Phi(0)^{3/2}} = \frac{\sqrt{2}}{2\pi} > 0, \tag{15}$$

$$\mu'(0) = \frac{d}{d\tau}\big[\phi(-\tau)\,\Phi(-\tau)^{-1}\big]_{\tau=0} = \frac{\phi(0)^2}{\Phi(0)^2} = \frac{2}{\pi} > 0, \tag{16}$$

$$D'(0) = \big[\mu(\tau) + \tau\mu'(\tau) - 2\,\mu(\tau)\,\mu'(\tau)\big]_{\tau=0} = \sqrt{\tfrac{2}{\pi}}\Big(1 - \tfrac{4}{\pi}\Big) < 0. \tag{17}$$

Since $C'(\tau) = -\frac{1}{2}\,(D(\tau) + \sigma^2)^{-3/2}\,D'(\tau)$, and from (14) and (17) we have

$$C'(0) = -\tfrac{1}{2}\,\underbrace{(D(0) + \sigma^2)^{-3/2}}_{>0}\,\underbrace{D'(0)}_{<0} > 0. \tag{18}$$

Combining (13), (14), (15), and (18) in the product rule for $A'(\tau)$ gives

$$A'(0) = a\Big[\underbrace{B'(0)\,C(0)}_{>0} + \underbrace{B(0)\,C'(0)}_{>0}\Big] > 0, \quad P_e'(0) = -\underbrace{\phi\big(-A(0)\big)}_{>0}\,\underbrace{A'(0)}_{>0} < 0. \tag{19}$$

This analysis, following the provided equations, demonstrates that under the AWGN model, initiating truncation (i.e. using a small threshold $\tau > 0$) strictly decreases the bit-error probability. Although this model is an idealization, it predicts the existence of an optimal threshold $\tau$, whose value is typically determined empirically, as discussed in Section B.2.

# B  Experimental Details and Additional Experiments

## B.1  Implementation Details

**Experiments on SD v2.1.**  In these experiments, we use Stable Diffusion v2.1 [3] to generate $512 \times 512$ images with a guidance scale of 7.5 and 50 DDIM denoising steps [2], into which we embed watermarks. For dwtDct, dwtDctSvd [18], and RivaGAN [19], we employ the implementation from the GitHub repository[4]. Stable Signature [22], TRW [10], GS [5], and PRCW [6] are run via their official GitHub codes. We fine-tune Stable Signature for 100 steps with a batch size of 4 on 400 images from the ImageNet2014 validation set [35]. Since TRW is a single-bit scheme, we evaluate it only in the detection setting, using its robust *Ring* mode in channel 3 as recommended. GS is used with its default paper settings, and PRCW is configured for 256 bit capacity. RivaGAN and Stable Signature retain their original capacities of 32 and 48 bits, respectively—higher capacities generally worsen extraction performance and image quality. However, none of these methods surpass T2SMark even under these generous capacity settings.

**Experiments on SD v3.5M.**  To assess the generalizability of our proposed T2SMark, we evaluate it alongside other inversion-based watermarking methods on the Stable Diffusion 3.5 Medium model (SD v3.5M) [8]. SD v3.5 adopts a Rectified Flow training objective [36], operates in a larger 16-channel latent space, and leverages the more scalable DiT transformer architecture. Because it also employs ODE-based sampling, inversion-based watermarking methods can be adapted to it directly. For robustness, we generate watermarked images for 500 prompts from the SDP training set with a guidance scale of 7.5; for image quality, we sample the COCO test set [28] with a guidance scale of 4.0 to emphasize diversity. All methods use 40 inference steps and 10 inversion steps. Embedding occurs in the 16-channel latent space: for TRW and T2SMark's first stage we use the first four channels; GS uses $f_{ch} = 4$, $f_h = 8$, and $f_w = 8$ (256-bit capacity); and PRCW and T2SMark both embed a 256-bit watermark. T2SMark further employs a 16-bit random key, as in our SD v2.1 setup.

## B.2  Parameter Selection

**Threshold $\tau$.**  We select $\tau$ by generating 256-bit, single-stage watermarks on 100 non-overlapping SDP prompts, applying distortions, and measuring bit accuracy. As shown in Table 8, bit accuracy first increases rapidly with $\tau$ and then slowly decreases. To balance robustness without over-constraining sampling, we choose $\tau = -\Phi^{-1}\left(\frac{4}{16}\right) \approx 0.674$.

Table 8: Bit accuracy under different truncation thresholds $\tau$.

| $\tau$ | 0 | $-\Phi^{-1}\left(\frac{7}{16}\right)$ | $-\Phi^{-1}\left(\frac{6}{16}\right)$ | $-\Phi^{-1}\left(\frac{5}{16}\right)$ | $-\Phi^{-1}\left(\frac{4}{16}\right)$ | $-\Phi^{-1}\left(\frac{3}{16}\right)$ | $-\Phi^{-1}\left(\frac{2}{16}\right)$ | $-\Phi^{-1}\left(\frac{1}{16}\right)$ |
|---|---|---|---|---|---|---|---|---|
| Bit Acc. | 0.9447 | 0.9828 | 0.9873 | 0.9860 | 0.9868 | 0.9855 | 0.9804 | 0.9600 |

Table 9: Post-distortion inversion MSE and detection accuracy (Det. Acc.) by channel.

| Channel | 0 | 1 | 2 | 3 |
|---|---|---|---|---|
| MSE $\downarrow$ | 0.8018 | **0.7815** | 0.8349 | 0.8352 |
| Det. Acc. $\downarrow$ | 0.578 | 0.612 | 0.633 | **0.553** |

**Embedding Channel of the Session Key.**  Different latent channels exhibit varying inversion errors and detectability. We measure the post-distortion inversion MSE and detection rate for channels 0–3 on the same dataset. As Table 9 shows, Channel 1 achieves the lowest MSE but is easily detected, while Channel 3 has the lowest detection rate but higher error. We therefore select **Channel 0** to balance undetectability and reconstruction fidelity.

---

[4]https://github.com/ShieldMnt/invisible-watermark

## B.3 Details about the $t$-test

For evaluating the CLIP score [30] and FID [31], we generated $n_s = n_0 = 10$ image sets for a two-sample $t$-test. We test the hypotheses

$$H_0 : \mu_s = \mu_0, \quad H_1 : \mu_s \neq \mu_0, \tag{20}$$

where $\mu_s$ and $\mu_0$ denote the mean metric values for watermarked and clean images, respectively. The $t$-statistic is computed as

$$t = \frac{|\mu_s - \mu_0|}{S^*\sqrt{\frac{1}{n_s} + \frac{1}{n_0}}}, \qquad S^* = \sqrt{\frac{(n_s - 1)S_s^2 + (n_0 - 1)S_0^2}{n_s + n_0 - 2}}, \tag{21}$$

where $S_s$ and $S_0$ are the sample standard deviations. A lower $t$-value supports $H_0$, while $t > t_{0.05, \, n_s + n_0 - 2} = t_{0.05, 18} \approx 2.101$ leads us to reject $H_0$ and conclude that watermarking significantly affects model performance. Otherwise, we consider the watermark to be distortion-free.

## B.4 Detailed Results of Robustness Evaluation

We report robustness evaluation results on SD v2.1 against various noise types in Table 10, with corresponding results on SD v3.5M in Table 11. T2SMark consistently outperforms all competing methods across nearly every distortion. Its single notable weakness is Gaussian noise, which likely arises because the inversion process is especially sensitive to this perturbation—an effect also observed in other inversion-based schemes [5, 10]. Cascading errors of the two-stage framework exacerbate this vulnerability, causing T2SMark to incur a greater performance decline under Gaussian noise.

Table 10: Detection (TPR) and traceability (bit accuracy) results for watermarking methods on SD v2.1 under various noise distortions. Values are shown as TPR/Bit Acc.

| Noise | dwtDct [18] | dwtDctSvd [18] | RivaGan [19] | StableSig [22] | TRW [10] | GS [5] | PRCW [6] | T2SMark |
|---|---|---|---|---|---|---|---|---|
| None | 0.922/0.8177 | **1.000**/0.9988 | 0.914/0.9823 | **1.000**/0.9981 | **1.000**/– | **1.000**/1.0000 | 1.000/0.6494 | **1.000/1.0000** |
| JPEG | 0.000/0.5015 | 0.002/0.5225 | 0.150/0.8091 | 0.384/0.7934 | 0.678/– | **1.000**/0.9794 | 0.448/0.5017 | **1.000/0.9901** |
| RandCr | 0.986/0.7675 | 0.998/0.7833 | 0.762/0.9476 | 0.994/0.9873 | 0.980/– | **1.000**/0.9421 | 0.078/0.4985 | **1.000/0.9935** |
| RandDr | 0.010/0.6012 | 0.000/0.6185 | 0.700/0.9352 | 0.984/0.9714 | 0.982/– | **1.000**/0.9156 | 0.052/0.5004 | **1.000/0.9895** |
| Resize | 0.000/0.5159 | 0.982/0.8301 | 0.740/0.9442 | 0.000/0.5130 | 0.998/– | **1.000**/0.9893 | 0.560/0.4983 | **1.000/0.9963** |
| GauBlur | 0.000/0.5029 | 0.352/0.6331 | 0.194/0.8171 | 0.000/0.4104 | 0.994/– | **1.000**/0.9661 | 0.070/0.4991 | **1.000/0.9890** |
| MedBlur | 0.000/0.5251 | 0.998/0.9113 | 0.780/0.9516 | 0.000/0.6511 | 0.982/– | **1.000**/0.9945 | 0.748/0.4997 | **1.000/0.9984** |
| GauNoise | 0.318/0.6236 | 0.850/0.7807 | 0.250/0.7948 | 0.458/0.7760 | 0.758/– | 0.986/**0.9014** | 0.054/0.4985 | **0.994**/0.8961 |
| S&PNoise | 0.114/0.5878 | 0.000/0.5168 | 0.082/0.7991 | 0.076/0.7020 | 0.948/– | **1.000**/0.9347 | 0.072/0.4992 | 0.992/**0.9456** |
| Brightness | 0.132/0.5439 | 0.060/0.5241 | 0.266/0.8009 | 0.866/0.9134 | 0.856/– | 0.992/0.9704 | 0.560/0.5259 | **0.994/0.9803** |
| Adv. (ave) | 0.173/0.5744 | 0.471/0.6800 | 0.432/0.8666 | 0.418/0.7462 | 0.907/– | **0.998**/0.9548 | 0.294/0.5024 | **0.998/0.9754** |

Table 11: Detection (TPR) and traceability (bit accuracy) results for watermarking methods on SD v3.5M under various noise distortions. Values are shown as TPR/Bit Acc.

| Noise | TRW[10] | GS [5] | PRCW [6] | T2SMark |
|---|---|---|---|---|
| None | 0.878 / - | **1.000** / 0.9994 | 0.998 / 0.9920 | **1.000 / 0.9998** |
| JPEG | 0.050 / - | **0.998** / 0.9868 | 0.312 / 0.5968 | **0.998 / 0.9955** |
| RandCr | 0.596 / - | **0.998** / 0.9894 | 0.482 / 0.6641 | **0.998 / 0.9989** |
| RandDr | 0.822 / - | 0.992 / 0.9694 | 0.474 / 0.6597 | **0.998 / 0.9955** |
| Resize | 0.030 / - | **0.998** / 0.9843 | 0.170 / 0.5596 | **0.998 / 0.9972** |
| GauBlur | 0.098 / - | 0.994 / 0.9487 | 0.004 / 0.5017 | **0.998 / 0.9790** |
| MedBlur | 0.004 / - | **0.998** / 0.9923 | 0.250 / 0.5993 | **0.998 / 0.9983** |
| GauNoise | 0.410 / - | **0.946 / 0.8832** | 0.038 / 0.5111 | 0.886 / 0.8629 |
| S&PNoise | 0.738 / - | **0.996** / 0.9528 | 0.026 / 0.5057 | **0.996 / 0.9664** |
| Brightness | 0.116 / - | 0.994 / 0.9898 | 0.756 / 0.8619 | **0.998 / 0.9978** |
| Adv. (ave) | 0.318 / - | **0.990** / 0.9663 | 0.279 / 0.6067 | 0.985 / **0.9768** |

## B.5 Results on Different Datasets

To avoid dataset-induced bias, we evaluate watermarking methods on two distinct datasets. Robustness is measured on the COCO dataset [28], while image quality and generation diversity are assessed on the SDP dataset. Since SDP provides no ground-truth references, the FID cannot be computed; therefore, we report only the CLIP score and LPIPS diversity. The results are summarized in Table 12.

Table 12: Additional evaluation results of watermarking methods, including detection (TPR), traceability (Bit Acc.), generation diversity, and visual quality (CLIP score). TPR and Bit Acc. shown as Clean/Adv. Standard errors and $t$-values are shown for the CLIP score.

| Method | TPR | Bit Acc. | Diversity ↑ | CLIP Score ($t \downarrow$) |
|---|---|---|---|---|
| SD v2.1 [3] | – | – | 0.6756 | 0.3357±.0008(–) |
| dwtDct [18] | 0.938/0.176 | 0.8493/0.5674 | – | 0.3347±.0007(2.9012) |
| dwtDctSvd [18] | **1.000**/0.472 | 0.9998/0.6739 | – | 0.3289±.0014(12.613) |
| RivaGAN [19] | 0.980/0.511 | 0.9951/0.8829 | – | 0.3324±.0009(8.5932) |
| StableSig [22] | 0.998/0.404 | 0.9965/0.7422 | 0.6667 | 0.3327±.0009(7.4739) |
| TRW [10] | **1.000**/0.892 | –/– | 0.6655 | 0.3393±.0007(10.139) |
| GS [5] | **1.000**/**0.997** | **1.0000**/0.9639 | 0.6156 | 0.3354±.0035(**0.2229**) |
| PRCW [6] | **1.000**/0.450 | 0.7956/0.5068 | **0.6747** | 0.3352±.0006(1.4729) |
| T2SMark | **1.000**/0.996 | **1.0000/0.9789** | 0.6746 | 0.3351±.0012(0.3634) |

## B.6  Details about Evaluation of Undetectability

To assess undetectability, we fine-tuned a pretrained ResNet-18 [32] classifier. For each inversion-based watermarking method and an unwatermarked Stable Diffusion baseline, we generated 8,500 images from the SDP training set—8,000 for training and 500 for testing. The semantic content was controlled with a fixed prompt set so that the classifier could not exploit content differences. All the watermark methods use a fixed key and watermark across images. Figures 6 and 7 respectively show the training loss and test accuracy over epochs on SD v2.1 and SD v3.5M. TRW [10] and GS [5] are easily detected, while PRCW [6] and T2SMark achieve only marginal detection accuracies just above 50%. Notably, PRCW is not completely immune to detection, with a test accuracy that is consistently greater than 50%. T2SMark performs better on SD v3.5M, indicating that the diffusion model itself critically impacts undetectability. Purely cryptographic undetectability may over-constrain the encoding and sacrifice robustness.

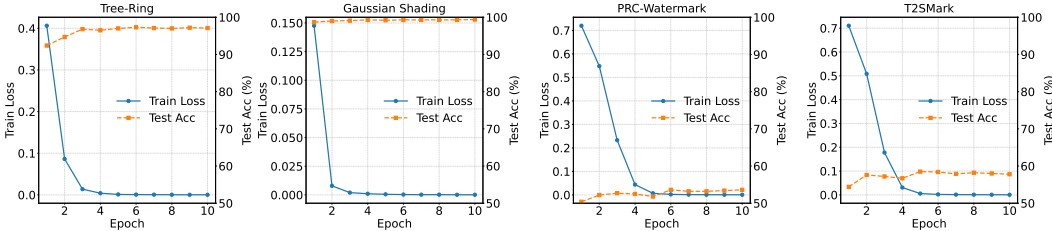

Figure 6: Training Loss and Test Accuracy over Epochs on SD v2.1.

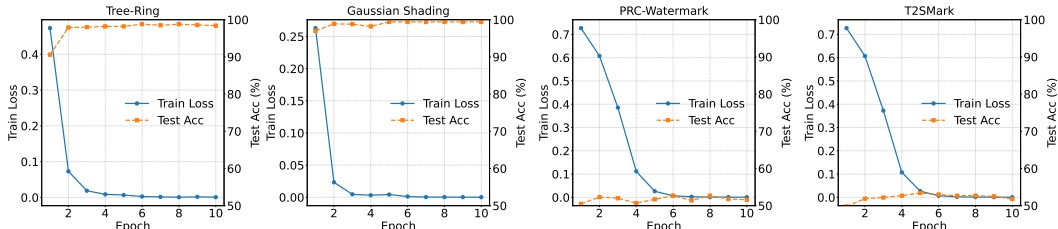

Figure 7: Training Loss and Test Accuracy over Epochs on SD v3.5M.

## B.7  Detailed Results of Ablation Studies

Tables 13, 14 and 15 present detailed ablation results for the inversion steps, watermark capacity and session key size. Within practical ranges, our method supports flexible adjustment of the embedding capacity—including the session key size—without altering any other parameters, making it well

suited for real-world deployment. Because diffusion models allow users to select image resolutions freely, our scheme naturally accommodates different resolutions without additional tuning, whereas methods such as GS [5] require modifying the diffusion factor, which can be cumbersome.

Under adversarial distortions, we observe a slight decline in bit accuracy as the number of inversion steps increases. We attribute this to complex interactions between certain perturbations, such as brightness changes, and the inversion process.

Table 13: Robustness evaluation of T2SMark across different inversion steps, shown as TPR/Bit Acc.

| Noise | 5 | 10 | 25 | 50 | 100 |
|---|---|---|---|---|---|
| None | 1.000/1.0000 | 1.000/1.0000 | 1.000/1.0000 | 1.000/1.0000 | 1.000/1.0000 |
| JPEG | 1.000/0.9886 | 1.000/0.9901 | 1.000/0.9904 | 1.000/0.9899 | 1.000/0.9879 |
| RandCr | 1.000/0.9899 | 1.000/0.9934 | 1.000/0.9899 | 1.000/0.9857 | 1.000/0.9816 |
| RandDr | 1.000/0.9851 | 1.000/0.9901 | 1.000/0.9873 | 1.000/0.9836 | 1.000/0.9791 |
| Resize | 1.000/0.9955 | 1.000/0.9963 | 1.000/0.9969 | 1.000/0.9968 | 1.000/0.9967 |
| GauBlur | 1.000/0.9835 | 1.000/0.9890 | 1.000/0.9878 | 1.000/0.9871 | 1.000/0.9867 |
| MedBlur | 1.000/0.9982 | 1.000/0.9984 | 1.000/0.9981 | 1.000/0.9981 | 1.000/0.9982 |
| GauNoise | 0.972/0.8849 | 0.986/0.9018 | 0.976/0.9077 | 0.992/0.9062 | 0.984/0.9034 |
| S&PNoise | 0.996/0.9373 | 0.996/0.9482 | 0.998/0.9530 | 0.998/0.9504 | 1.000/0.9535 |
| Brightness | 0.994/0.9741 | 0.992/0.9803 | 0.990/0.9760 | 0.990/0.9728 | 0.990/0.9698 |
| Adv. (ave) | 0.996/0.9708 | 0.997/0.9764 | 0.996/0.9764 | 0.998/0.9745 | 0.997/0.9730 |

Table 14: Bit accuracy of T2SMark watermark extraction across different embedding capacities.

| Noise | 256 | 384 | 512 | 768 | 1024 |
|---|---|---|---|---|---|
| None | 1.0000 | 0.9999 | 0.9999 | 0.9992 | 0.9968 |
| JPEG | 0.9901 | 0.9766 | 0.9648 | 0.9381 | 0.9107 |
| RandCr | 0.9935 | 0.9814 | 0.9630 | 0.9313 | 0.8936 |
| RandDr | 0.9895 | 0.9774 | 0.9488 | 0.9236 | 0.8425 |
| Resize | 0.9963 | 0.9900 | 0.9816 | 0.9585 | 0.9332 |
| GauBlur | 0.9889 | 0.9700 | 0.9524 | 0.9140 | 0.8816 |
| MedBlur | 0.9984 | 0.9939 | 0.9878 | 0.9692 | 0.9482 |
| GauNoise | 0.8961 | 0.8691 | 0.8487 | 0.8123 | 0.7744 |
| S&PNoise | 0.9456 | 0.9147 | 0.8960 | 0.8547 | 0.8205 |
| Brightness | 0.9803 | 0.9628 | 0.9509 | 0.9287 | 0.9056 |
| Adv. (ave) | 0.9754 | 0.9595 | 0.9438 | 0.9145 | 0.8789 |

Table 15: Robustness performance of T2SMark watermark with different random key sizes, shown as TPR/Bit Acc.

| Noise | 8 | 16 | 24 | 32 |
|---|---|---|---|---|
| None | 1.000/0.9999 | 1.000/1.0000 | 1.000/1.0000 | 1.000/0.9999 |
| JPEG | 1.000/0.9900 | 1.000/0.9901 | 1.000/0.9898 | 1.000/0.9730 |
| RandCr | 1.000/0.9930 | 1.000/0.9935 | 1.000/0.9937 | 1.000/0.9718 |
| RandDr | 1.000/0.9897 | 1.000/0.9895 | 1.000/0.9880 | 1.000/0.9787 |
| Resize | 1.000/0.9966 | 1.000/0.9963 | 1.000/0.9973 | 1.000/0.9930 |
| GauBlur | 1.000/0.9868 | 1.000/0.9890 | 1.000/0.9862 | 1.000/0.9760 |
| MedBlur | 1.000/0.9980 | 1.000/0.9984 | 1.000/0.9984 | 1.000/0.9982 |
| GauNoise | 0.994/0.9113 | 0.994/0.8961 | 0.982/0.8629 | 0.940/0.8172 |
| S&PNoise | 0.998/0.9512 | 0.992/0.9456 | 1.000/0.9340 | 0.998/0.8802 |
| Brightness | 0.990/0.9803 | 0.992/0.9803 | 0.986/0.9680 | 0.984/0.9443 |
| Adv. (ave) | 0.998/0.9774 | 0.998/0.9754 | 0.996/0.9687 | 0.991/0.9481 |

## B.8 Analysis of Computational Overhead

We conducted a timing comparison for all watermark methods. Experiments were run on a system equipped with an AMD EPYC 7662 Processor and an NVIDIA RTX A6000 GPU. All inversion-based methods utilized FP16 precision and performed 10 inversion steps on an SD v2.1 model [3].

Table 16: Timing comparison of various watermarking methods. Embedding and verification times are measured in seconds (s).

| Method | Embedding Time (s) | Verification Time (s) |
|---|---|---|
| dwtDct[18] | 0.047 | 0.030 |
| dwtDctSvd[18] | 0.113 | 0.065 |
| rivaGan[19] | 1.327 | 1.087 |
| StableSig[22] | 0.000 | 0.004 |
| TRW[10] | 0.001 | 0.437 |
| GS[5] | 2.253 | 0.473 |
| PRCW[6] | 0.006 | 0.821 |
| T2SMark | 0.024 | 0.420 |

### B.9 Impact of the Two-Stage Framework

To assess the impact of the two-stage framework, we evaluate T2SMark both with and without it. All experiments use Stable Diffusion v2.1 [3] under the same settings as the main paper. For robustness testing, we sample 500 prompts from the Stable-Diffusion-Prompts training split; for generation diversity, we use 1,000 prompts from the same dataset and report LPIPS scores [29]. Results are shown in Table 17.

Enabling the two-stage framework reduces adversarial bit accuracy from 0.9868 to 0.9754, since reserving latent dimensions to encrypt the session key cuts redundancy and causes cascading errors—any mistake in decoding the session key invalidates the second-stage decoding. However, it substantially increases generation diversity, confirming its importance for balancing robustness and diversity. In contrast, without two-stage encryption, T2SMark suffers from the same fixed-codeword limitation as Gaussian Shading [5] and overconcentrates energy in the high-energy tail region, whose position in the latent vector is entirely key-dependent , significantly reducing diversity.

Table 17: Performance of T2SMark both with and without the two-stage framework.

|  | Bit Acc. (Clean/Adv.) | Diversity ↑ |
| --- | --- | --- |
| w/o two-stage | 1.0000/0.9868 | 0.5689 |
| w/ two-stage | 1.0000/0.9754 | 0.6746 |

## C  Visual Results

**Generation Diversity.**    Figure 8, 9, 10, and 11 visually compare the generation diversity achieved by various watermarking methods on SD v2.1 and SD v3.5M. Images produced with Gaussian Shading [5], for example, display a highly consistent layout, indicating limited diversity.

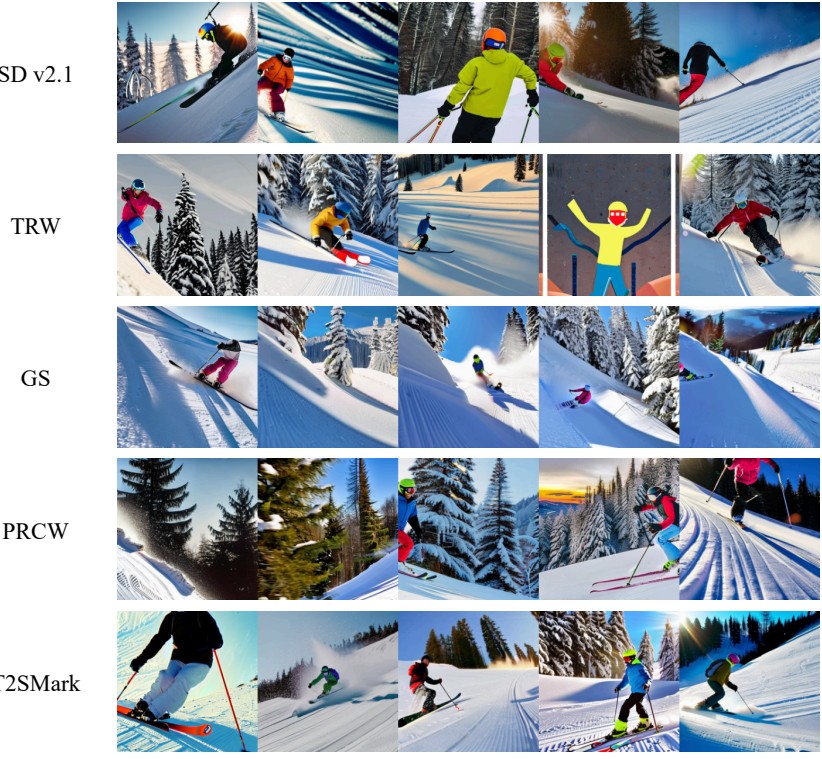

Figure 8: Diversity of images generated by different watermarking methods via SD v2.1 (guidance scale = 7.5, 50 inference steps) on the prompt *"A guy is having fun skiing down the slope of the hill."*

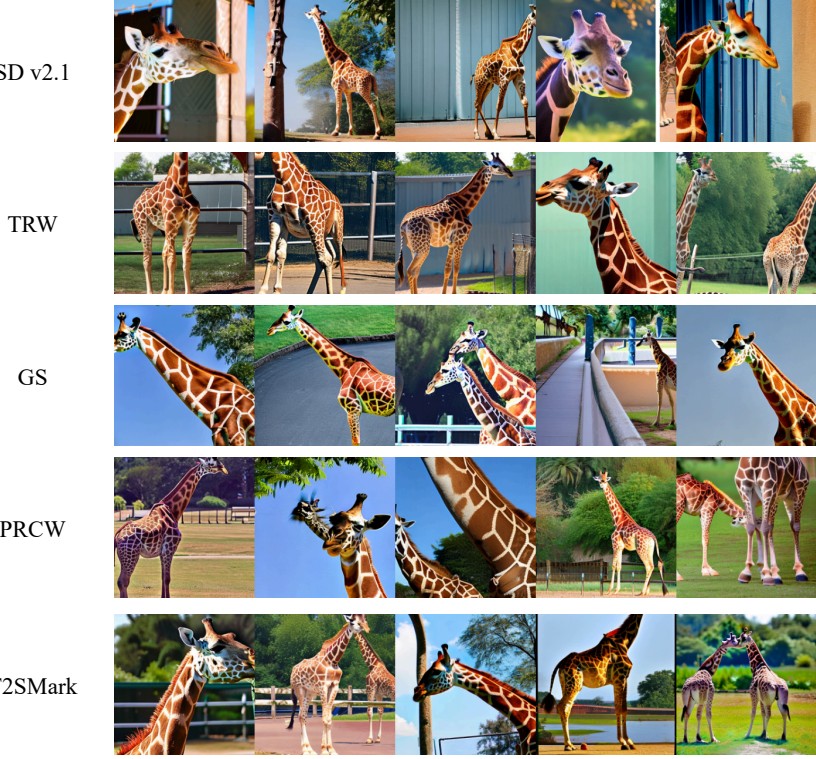

Figure 9: Diversity of images generated by different watermarking methods via SD v2.1 (guidance scale = 7.5, 50 inference steps) on the prompt *"Two giraffes in a zoo enjoy a walk and a snack."*

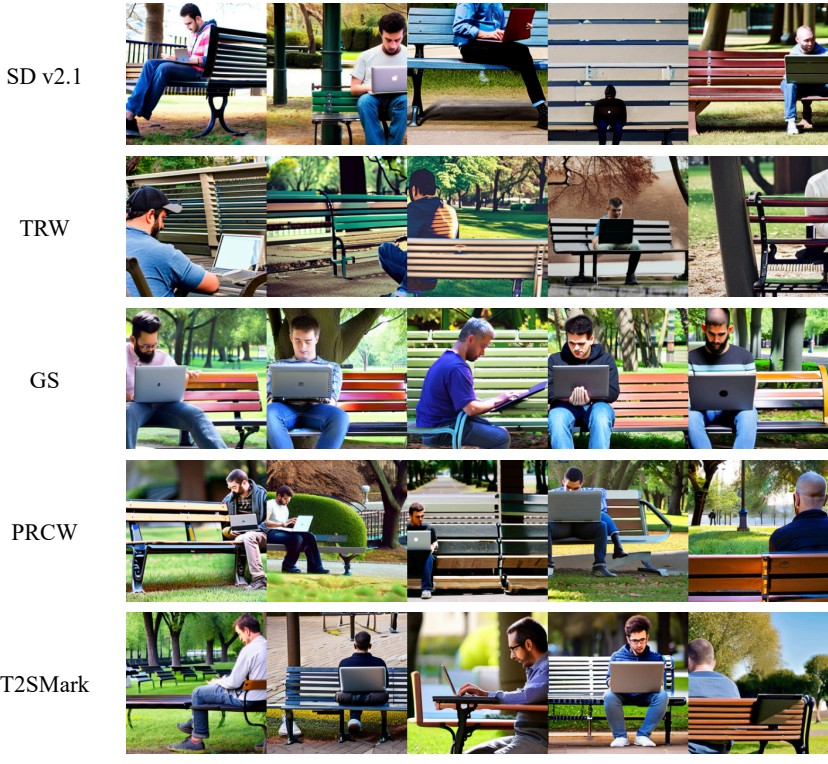

Figure 10: Diversity of images generated by different watermarking methods via SD v2.1 (guidance scale = 7.5, 50 inference steps) on the prompt *"A man uses his computer on a park bench."*

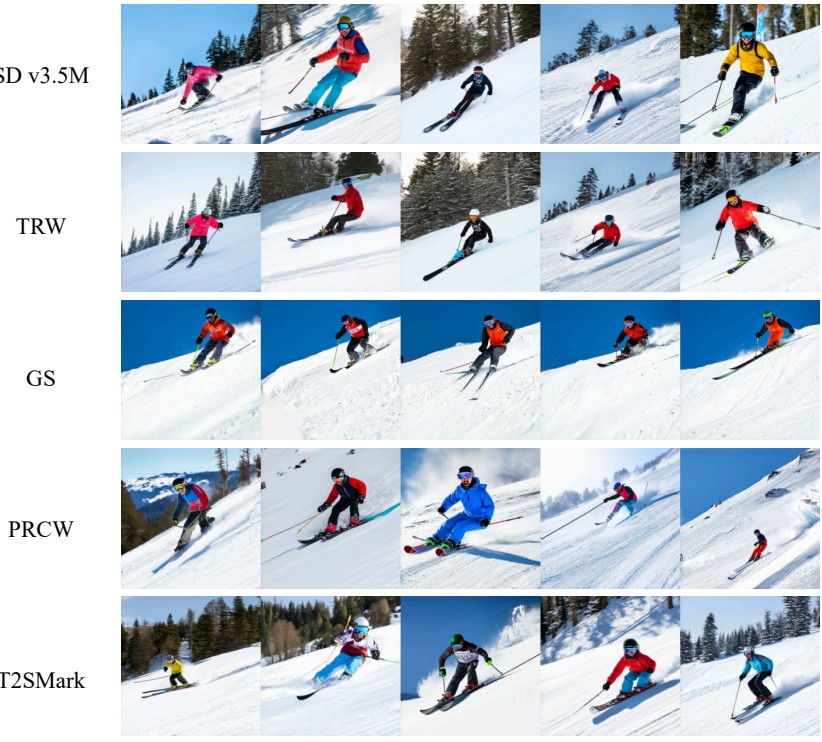

Figure 11: Diversity of images generated by different watermarking methods using SD v3.5M (guidance scale = 4.0, 40 inference steps) on the prompt *"A guy is having fun skiing down the slope of the hill."*

