# OpenReview forum: "T2SMark: Balancing Robustness and Diversity in Noise-as-Watermark for Diffusion Models"
_NeurIPS.cc/2025/Conference — NeurIPS 2025 poster_

### Official Review · Reviewer_jfmo · 2025-06-29

**Clarity:** 3
**Significance:** 3
**Originality:** 3
**Rating:** 5
**Confidence:** 3

**Summary:**

This paper proposes T2SMark, a novel two-stage watermarking framework for diffusion models that balances robustness and generation diversity. The method embeds watermark bits exclusively in reliable tail regions of Gaussian noise while randomly sampling the central zone to achieve robustness and diversity. A two-stage key hierarchy (master key + session key) injecting controlled randomness to further preserve diversity. Extensive experiments demonstrate superior watermark traceability, diversity, and minimal visual quality degradation.

**Questions:**

1. Fundamentally, this work is an extension of the tree-ring watermarking approach, which also modifies the initial noise. Similar methods (tree-ring, Gaussian shading, and PRC) inherently produce images that differ from the original, unwatermarked generations, leading to potentially uncontrolled diversity variations. Newer methods, such as ROBIN [1] and ZoDiac [2], enable watermark embedding during the generation process itself, achieving outputs that are visually indistinguishable from their watermark-free counterparts. The paper, however, appears to overlook direct comparisons with these relevant recent techniques.

[1] ROBIN: Robust and Invisible Watermarks for Diffusion Models with Adversarial Optimization. 2024 NeurIPS.

[2] Attack-Resilient Image Watermarking Using Stable Diffusion. 2024 NeurIPS.

2. Undetectability evaluation assumes fixed watermark and key: This is an unrealistic assumption in real-world settings, where keys vary per user. The classifier’s performance might change substantially in such settings, especially since randomness is a core contribution of T2SMark. Further evaluation with randomized watermark instances would strengthen the claim of imperceptibility.

3. Error propagation can be observed in two-stage decoding under strong noise (Tab.11 GauNoise). Robustness could be improved via error-correcting codes?

**Ethical Concerns:**

["NO or VERY MINOR ethics concerns only"]

**Final Justification:**

See Official Comment.

**Limitations:**

yes

**Paper Formatting Concerns:**

No formatting concerns

**Quality:**

3

**Strengths And Weaknesses:**

### Strengths：
1.The conflict between robustness and diversity in watermarking diffusion outputs is well-articulated and timely, given the increasing deployment of diffusion models.

2.The two-stage key hierarchy decouples randomness control from watermark encoding, ensuring diversity without sacrificing security.

3.The paper is well-structured and technically accessible.


### Weaknesses：
1. The watermarking pipeline by modifying the initial noise in diffusion models is relatively common. However, such modification can be substantial and may potentially affect the model's applicability in controllable generation methods that rely on initial noise characteristics, such as Golden noise [1] and InitNO [2].

[1] Golden Noise for Diffusion Models: A Learning Framework. 2411.09502 arXiv.

[2] InitNO: Boosting Text-to-Image Diffusion Models via Initial Noise Optimization. 2024 CVPR.

2. While the paper emphasizes improvements in diversity and provides quantitative results (e.g., LPIPS), the visual evidence is limited. Only one example visualization (Fig. 8 in the Appendix) under a single prompt is provided. Comparisons with more baseline methods through visual results are notably lacking.

3. The paper lacks a comparison of the time overhead, specifically in terms of watermark embedding and verification times, with other existing schemes.

4. The decision to embed the session key in channel 0 (based on inversion error and detectability) appears empirically driven, but the rationale may not generalize across models or sampling resolutions.

5. While multiple attack scenarios are tested, inclusion of rotation would strengthen claims given its prevalence in image manipulations.

---

> ### Author Rebuttal · Authors · 2025-07-31
>
> Thank you for your insightful comments on our work. Our responses to each of the points you raised are detailed below.
>
> ---
>
> ### **Re: Impact on applicability of controllable generation methods (Weakness 1)**
>
> Thank you for pointing it out! We will add a discussion on this to Section 5. However, not all controllable generation methods are affected. Our NaW method does not modify already sampled initial noise; instead, it replaces the original initial noise sampling logic with a pseudo-random one. The sampled noise still adheres to the original distribution and is indistinguishable from standard noise without the correct key.
>
> To clarify, the following discussion is structured by the type of controllable generation method:
>
> 1. **Methods that modify the initial noise sampling logic (e.g., require specific noise patterns):** Our watermarking method would conflict with these, as we already replace the default sampling.
> 2. **Methods that require only minor modifications to the initial noise:** Our method remains applicable. For instance, with Golden Noise, its perturbation to the initial noise is minimal, far less than typical image distortions or inversion errors.
> 3. **Methods that irreversibly alter the denoising process:** Our method is incompatible here. InitNO, for example, performs gradient-based optimization at various denoising timesteps, which compromises the determinism of ODE-based sampling. This is why it cannot adapt to our method, as mentioned in Section 5. If an inversion method corresponding to InitNO's denoising process could be found, we believe our approach would still be applicable.
>
> ---
>
> ### **Re: Visual evidence is limited (Weakness 2)**
>
> Current NeurIPS 2025 policy does not allow us to provide additional visual evidence directly within the rebuttal or discussion phase. However, we are fully committed to including more example images in Appendix C if the paper is accepted, to clearly showcase both diversity and maintained image quality.
>
> As compensation and to address your concern regarding diversity, we conducted a **user study**. We randomly selected 20 prompts for each of 25 participants. For every prompt, we presented a set of five images generated by different watermarking methods and also a non-watermarked baseline. Participants were then asked to choose the set they perceived as most diverse. Our results, presented below, largely align with our quantitative diversity measurements in the paper, which we hope alleviates your concerns:
>
> | Method | SD v2.1 | StableSig | TRW | GS | PRCW | T2SMark |
> | --- | --- | --- | --- | --- | --- | --- |
> | Win Rate | 0.174 | 0.140 | 0.128 | 0.112 | 0.240 | 0.206 |
>
> ---
>
> ### **Re: Lack of comparison of the time (Weakness 3)**
>
> We conducted a timing comparison for all inversion-based methods. Experiments were run on a system equipped with an AMD EPYC 7662 Processor and an NVIDIA RTX A6000 GPU. All inversion-based methods utilized FP16 precision and performed 10 inversion steps on an SD v2.1 model.
>
> |  | Embedding Time(s) | Verification Time(s) |
> | --- | --- | --- |
> | dwtDct | 0.047  | ***0.030*** |
> | dwtDctSvd | 0.113 | 0.065 |
> | rivaGan | 1.327 | 1.087 |
> | StableSig | **0.000** | **0.004** |
> | TRW | ***0.001*** | 0.437 |
> | GS | 2.253 | 0.473 |
> | PRCW | 0.006 | 0.821 |
> | T2SMark | 0.024 | 0.420 |
>
> ---
>
> ### **Re: The Empirical decision of the channel embedding channel of session key (Weakness 4)**
>
> The choice of channel is quite flexible, and Table 9 shows that performance differences are minimal. In our generalization experiments with SD v3.5M (Section 4.4), we did not specifically select channels, and its latent space has 16 channels, which is different from SD v2.1's 4 channels. For simplicity, we chose the first four channels to embed the session key and still achieved considerable results (see Table 3). More detailed experimental settings can be found in Appendix B.1.
>
> ---
>
> ### **Re: Rotation (Weakness 5)**
>
> In fact, rotation presents a notable challenge for NaW methods like GS, PRCW, and T2SMark since their designs lack explicit mechanisms to resist it, highlighting a key area for research. We will clearly point out this limitation in the paper.
>
> The recent ICML 2025 paper, GaussMarker [1], offers a promising solution. They developed a novel dual-domain watermark, embedding NaW in the spatial domain and TRW in the frequency domain. They also train a network to recover rotated images for the spatial watermark. Since they successfully used GS as their spatial watermark, we are confident that integrating our method in place of their spatial domain watermark would similarly achieve strong resistance against rotation.
>
> [1] GaussMarker: Robust Dual-Domain Watermark for Diffusion Models, ICML 2025.
>
> ---
>
> ### **Re: Comparisons with ROBIN and ZoDiac (Question 1)**
>
> We acknowledge the impressive fidelity to original images and competitive robustness of methods like ROBIN and ZoDiac. Thank you for these helpful references and we will include a discussion of them in the related work section.
>
> However, these are 0-bit watermarking schemes, designed solely for detection, and they require gradient-based optimization. In contrast, by not needing to be strictly identical to the original image, training-free NaW methods can fully leverage the initial noise space to achieve a 256-bit capacity for traceability. Given these significant differences in design philosophy, a direct comparison would be unintuitive.
>
> ---
>
> ### **Re: Assumptions of the undetectability evaluation (Question 2)**
>
> We believe there might be a misunderstanding regarding our key assumptions. Our method employs two types of keys: a **master key** and a **session key**. The session key varies with every generation, while the master key remains fixed. We contend that the master key should not vary per user. In our threat model (Section 3.1), to trace the origin of an image, we extract the watermark content, which depends on the master key for decoding. If the master key varied per user, we would paradoxically need to know who generated the image *before* decoding the watermark, creating a contradiction.
>
> We suspect your concern might pertain to the watermark content varying per user, which is indeed the case. However, we consider a more challenging scenario: an attacker seeking to train a classifier would most conveniently obtain sufficient watermarked image samples from a single user's API. In such a scenario, both the master key and watermark content would be fixed, providing the classifier with more consistent information to learn common features. This makes it a more challenging test for watermark undetectability. We will clarify this point clearly in the paper.
>
> Regarding scenarios with randomized watermark instances (varying watermark content), GS, PRC, and T2SMark achieve comparable performance in our experimental setup, all hovering around 50% accuracy. This indicates that this scenario is indeed less challenging.
>
> ---
>
> ### **Re: Vulnerability to Gaussian noise and corresponding solutions (Question 3)**
>
> That is a good suggestion! You are right, error-correcting codes could indeed boost our method's robustness. We attempted to incorporate a 1-bit error-correcting Hamming code in the first stage. This increased bit accuracy under Gaussian noise from 0.8982 to 0.9068, while keeping the session key length constant. Multi-bit error-correcting codes like LDPC, might yield better results, but both ECC encoding and our watermarking method require redundancy. Given the limited capacity of the latent space, we found it challenging to balance the allocation of this redundancy between the two.
>
> We observed that removing Gaussian noise in the pixel domain *before* inversion appears to be a simpler and more effective strategy than applying error correction *after* inversion. Specifically, we found that independent additive noise, including Gaussian noise, is surprisingly easy for a model to distinguish. Plus, removing this noise using traditional filtering methods before inversion is highly effective. To illustrate this, we trained a simple four-layer CNN for an experiment. Here are the results:
>
> |  | Clean (TPR/Bit Acc) | GauNoise (TPR/Bit Acc) | S&P (TPR/Bit Acc) | Other Noise (TPR/Bit Acc) | Adv ave. (TPR/Bit Acc) |
> | --- | --- | --- | --- | --- | --- |
> | Without Classifier&Filter | 1.000/1.000 | 0.988/0.8982 | 0.998/0.9487 | 0.999/0.9913 | 0.998/0.9762 |
> | With Classifier&Filter | 1.000/1.000 | 0.996/0.9495 | 1.000/0.9971 | 0.999/0.9909 | 0.999/0.9870 |
>
> These results show that this plug-and-play module improves performance under Gaussian and Salt-and-Pepper noise, with only a marginal effect on other distortions. In fact, our classifier successfully detected the target noise with a 94.9% probability, while maintaining a low false positive rate of only 2.6%.

---

> > ### Comment · Reviewer_jfmo · 2025-08-04
> >
> > I appreciate the authors' detailed response, which has addressed most of my concerns. Accordingly, I will raise my score by one point. I strongly encourage the authors to incorporate key elements of their rebuttal into the manuscript—particularly the analysis of computational overhead and comprehensive robustness evaluation and discussion (including geometric transformations such as rotation). These aspects are critical for evaluating watermarking techniques and will strengthen the paper’s impact.

---

> > > ### Author Response · Authors · 2025-08-04
> > > **Thanks for your comment!**
> > >
> > > Dear Reviewer jfmo, thank you for your positive acknowledgment of our work. We are very pleased that we could address your concerns. We will incorporate the relevant results and discussions from our rebuttal directly into the final manuscript. Thank you for helping to improve our work again!

---

### Official Review · Reviewer_qjHw · 2025-06-29

**Clarity:** 2
**Significance:** 2
**Originality:** 2
**Rating:** 5
**Confidence:** 3

**Summary:**

The paper addresses the challenge of balancing robustness and generation diversity in noise-as-watermark (NaW) methods for diffusion models. Existing NaW approaches often encode watermarks in sensitive regions of the Gaussian noise space, which compromises either the watermark robustness or the diversity of generated images. To address this, the authors propose T2SMark, a two-stage encoding framework that introduces (1) Tail-Truncated Sampling (TTS) restricts watermark embedding to the more stable tails of the Gaussian distribution, and (2) a Two-Stage secure encoding scheme using random session keys.

**Questions:**

-	LPIPS may not be ideal for measuring diversity. While LPIPS reflects perceptual difference, it may not fully capture diversity. Other metrics like Inception Score and CLIP Score may offer complementary insights.

**Ethical Concerns:**

["NO or VERY MINOR ethics concerns only"]

**Final Justification:**

The author's rebuttal has addressed my concerns. However, since it is not closely related to my research topic, my confidence is 3.  Thanks to ACs, Reviewers, and Authors.

**Limitations:**

yes

**Quality:**

2

**Strengths And Weaknesses:**

Strengths:

-	Compatible with existing diffusion models. Similar to other NaW approaches, T2SMark does not require any architectural modifications or fine-tuning, making it practical and plug-and-play.
-	Tail-Truncated Sampling (TTS) is insightful and effective for the balance of robustness and diversity.

Weaknesses:

-	The undecided zone may affect the length of the injectable watermark and limit the watermarking capacity.
-	Evaluation lacks direct fidelity metrics. Using FID between two unpaired sets is indirect. Pairwise metrics like PSNR between clean and watermarked images would more accurately quantify distortion.
-	Figure 1 lacks quantitative support. The current illustration is too abstract. Including numerical comparisons would make the visual argument more credible.

---

> ### Author Rebuttal · Authors · 2025-07-31
>
> Thank you for your helpful feedback. We provide our responses to each point below.
>
> ---
>
> ### **Re: Reduction of the capacity (Weakness 1)**
>
> Our method’s moderation of theoretical capacity is a deliberate design choice, trading a largely academic ceiling, which no current method can practically approach, for substantial gains in real-world robustness. For example, when SD v2.1 generates images at a resolution of 512x512 pixels, the theoretical capacity limit for other NaW methods is determined by the latent space size, specifically 4×64×64 = 16,384 bits. Under our default settings, TTS reduces this upper limit to 8,192 bits by excluding those easy-flipping points. However, no existing method can maintain acceptable performance at such large capacities. In practical scenarios, methods typically operate at significantly lower capacities to ensure robustness. When compared at equivalent and practical capacities, our method consistently demonstrates superior robustness.
>
> ---
>
> ### **Re: Evaluation lacks direct fidelity metrics (Weakness 2)**
>
> We would like to clarify that our FID is computed between 1,000 generated images and 1,000 corresponding real images, ensuring semantic alignment.
>
> PSNR is a relative metric that measures deviation from an original image. As a NaW method, we directly encode the watermark into the initial noise, meaning there is no strict "original image" for direct comparison. Instead, we assess image quality by evaluating the overall quality of the generated content. In this paper, we generated a total of 10,000 images, organized into 10 groups. Similar to GS, we performed t-tests on CLIP and FID scores against unwatermarked images for all methods. A smaller t-value indicates greater indistinguishability from unwatermarked images. The t-test results are provided in Table 1 and Table 3, with further details in Appendix B.3. We also offer results on additional datasets in Appendix B.5, which we believe is sufficiently convincing.
>
> ---
>
> ### **Re: Figure 1 lacks quantitative support (Weakness 3)**
>
> Thank you for pointing it out! Our original intention was to visually convey how our method achieves a superior balance compared to two SOTA approaches, but the absence of specific quantitative data can indeed be misleading. We have modified the figure following your suggestion. Unfortunately, due to conference regulations, we cannot provide it immediately. We commit to updating this illustration in the camera-ready version if our paper is accepted.
>
> ---
>
> ### **Re: Further estimation of the diversity (Question 1)**
>
> In the context of measuring diversity, we employ LPIPS in a non-standard manner. While LPIPS usually measures the direct visual distance between two images (where a smaller value means more similarity and often better visual quality), we calculate it differently for diversity. For $P$ prompts, and for each prompt generating $N$ images, we compute the visual distance for all $C_N^2$ possible pairs. We then average these distances for each prompt and finally average across all $P$ prompts, formally:
>
> $D = \frac{1}{P}\sum_{i=1}^{P} \frac{1}{C_N^2} \sum_{j,k,1\leq j<k \leq N} d(X_{i,j}, X_{i,k})$
>
> If the average distance between multiple images generated for the same prompt is small, we consider them similar and thus lacking diversity. Conversely, a larger average distance indicates better diversity. The function d can also be replaced by other distance metrics like CLIP-Image similarity, where a larger value indicates more similarity; therefore, a smaller value would indicate better diversity. Specific results are in the table below.
>
> We also conducted a user study. We randomly selected 20 prompts for each of 25 participants. For every prompt, we provided a set of five images generated by different watermarking methods and a non-watermarked baseline. Participants were asked to choose the set they perceived as most diverse. Our results, consistent with the quantitative diversity measurements in the paper, are shown below, and we hope this addresses your concerns.
>
> Our conclusions on diversity align with previous work. PRC, as a cryptographically undetectable watermark, exhibits diversity nearly comparable to unwatermarked images, while GS shows the lowest diversity due to its fixed key. Our undetectability experiments (Section 4.3) further support this, as the classifier easily distinguishes between GS and unwatermarked images by learning common features.
>
> Inception Score calculates class diversity within an image set, making it more suitable for unconstrained generation. Since we use fixed prompts for each method, Inception Score appears unable to reflect the specific diversity in our context.
>
> |  | Diversity-LPIPS-SDP ↑ | Diversity-LPIPS-COCO ↑ | Diversity-CLIPI-SDP ↓ | Diversity-CLIPI-COCO ↓ | User-Study-WR ↑ |
> | --- | --- | --- | --- | --- | --- |
> | SD v2.1 | 0.6756 | 0.7072 | 0.7085 | 0.6949 | 0.174 |
> | StableSig | 0.6667 | 0.6917 | **0.7034** | 0.7051 | 0.140 |
> | TRW | 0.6655 | 0.6943 | 0.7152 | 0.6939 | 0.128 |
> | GS | 0.6156 | 0.6446 | 0.7364 | 0.7123 | 0.112 |
> | PRCW | **0.6747** | **0.7074** | ***0.7095*** | **0.6913** | **0.240** |
> | T2SMark | ***0.6746*** | ***0.7069*** | 0.7112 | ***0.6939*** | ***0.206*** |

---

> > ### Comment · Reviewer_qjHw · 2025-08-04
> > **Comment to Authors**
> >
> > Thanks for the rebuttal. The authors have addressed all of my concerns. I expect a more objective quantitative analysis in Figure 1. I will raise my final score.

---

> > > ### Author Response · Authors · 2025-08-04
> > > **Thanks for your comment!**
> > >
> > > Dear Reviewer qjHw, thanks for recognizing our responses. We are happy that our response has addressed your concerns. We will revise Figure 1 in the final manuscript to include quantitative results, drawing from the results in Table 1, to ensure a more objective analysis. Thank you again for your thoughtful review and support.

---

### Official Review · Reviewer_bX8p · 2025-06-30

**Clarity:** 3
**Significance:** 3
**Originality:** 2
**Rating:** 4
**Confidence:** 4

**Summary:**

The paper proposes T2SMark , a watermarking framework for diffusion models that balances robustness against distortions and enhances generation diversity. It introduces a two-stage approach with a tailored inversion process to embed watermarks and a tail-truncated sampling (TTS) strategy to enhance robustness.

**Questions:**

1. Following the weakness 1. Besides the TTS sampling, is there any other contributions make T2SMark significantly outperform the GS baseline?
2. Following the weakness 2. Can the authors provide a theoretical analysis of TTS sampling and why it makes watermarking more robust and diverse? How is the threshold $\tau$ calculated, or is it determined empirically?
3. Following the weakness 3. Despite the TTS mechanism, why does T2SMark remain vulnerable to Gaussian noise? Could this be mitigated via robust optimization during inversion?
4. The diversity is measured by the LPIPS metrics. While LPIPS is commonly used for image perceptual quality. I have a concern about this metric for its effectiveness in representing diversity. Can the authors provide more evidence to support the appropriate usage of LPIPS for diversity? To further estimate the diversity, I believe authors should show more visualizations and even user study to support their claim.
5. The GS baseline can keep the watermarked results similar to the original non-watermarked generated results, and it is also able to show diversity to some extent. Can the T2SMark method also preserve the similarity between the watermarked results and non-watermarked model results? So that users can flexibly watermark their generated results.
6. Can the proposed method be extended to the Diffusion Transformer method?

**Ethical Concerns:**

["NO or VERY MINOR ethics concerns only"]

**Final Justification:**

See the comments.

**Limitations:**

yes

**Quality:**

2

**Strengths And Weaknesses:**

Strengths

1. T2SMark introduces TTS, which partitions the Gaussian noise distribution into three regions bit-0, bit-1, and an undecided central zone. By embedding information exclusively in the reliable tail regions and randomly sampling the central area, the method reduces sign errors caused by noise perturbations while preserving the latent distribution.

2. The two-stage framework with hierarchical encryption structure to randomize bit messages, ensuring diversity in generated outputs without compromising security.

Weakness

1. Similar to Gaussian Shading (GS). T2SMark proposes a similar watermarking paradigm to the GS method. They all use a master key to generate a session key and then encrypt it into the noise vector. The watermark decoding all relies on the DDIM. There is a lack of comparison with the GS baseline to highlight the contribution of the T2SMark from the methodology perspective.

2. Limited theoretical insights. The paper focuses on empirical validation with minimal theoretical analysis of the TTS mechanism or robustness guarantees. The TTS sampling also relies on the pre-defined threshold $\tau$, which can be case-by-case sensitive.

3. Vulnerability to Gaussian noise: The inversion process remains sensitive to this distortion, a known issue in prior inversion-based methods. The authors acknowledge but do not resolve this.

---

> ### Author Rebuttal · Authors · 2025-07-31
>
> Thank you for your constructive feedback and detailed evaluation. We have addressed each of the points raised in our responses below.
>
> ---
>
> ### **Re: Contributions besides the TTS sampling (Question 1 & Weakness 1)**
>
> T2SMark's improvements go beyond TTS sampling. T2SMark enhances watermark randomness through its two-stage framework. We also would like to make it clearer: our session key is not generated from the master key; instead, it is independently randomized in every generation. This leads to better generation diversity compared to GS (quantitative data is in Table 1; furthermore, in Figure 8, GS-generated images consistently show a black shadow in the bottom-left region, a problem not seen with other methods). In terms of security, T2SMark is also notably harder to detect than GS (Table 2).
>
> ---
>
> ### **Re: Theoretical analysis of TTS sampling (Question 2 & Weakness 2)**
>
> Due to main paper page limitations, our detailed theoretical analysis of TTS is provided in Appendix A, explaining why TTS leads to stronger robustness.
>
> Regarding diversity, while the random region in TTS does offer some compensatory effects, the primary contribution comes from our two-stage framework. This is because the codewords reside in the tail regions, where they possess higher energy. For instance, a 16-bit session key can still provide 65,536 random variations for the codeword, even when the master key and watermark remain fixed. An ablation study on the two-stage framework, demonstrating its considerable improvement in diversity, is included in our supplementary materials.
>
> Concerning the threshold determination, we set it empirically rather than by calculation (Appendix B.2). This approach is taken because real-world noise environments can be complex and may not perfectly align with theoretical assumptions. Specifically, we iterated through various Gaussian distribution quantiles and chose a value that demonstrates favorable performance. Moreover, the truncation threshold (τ) performs poorly only at extreme values and maintains similar performance across most of its range, demonstrating robustness. Our generalization experiments (Section 4.4) further support this: we achieved SOTA performance (Table 3) on new architectures without any empirical hyperparameter re-tuning.
>
> ---
>
> ### **Re: Vulnerability to Gaussian Noise (Question 3 & Weakness 3)**
>
> TTS is a general enhancement that improves watermark performance across all distortions. Gaussian noise impacts our method more significantly due to our two-stage framework, where the session key in the first stage must be perfectly recovered; even a single bit error can lead to decoding failure.
>
> We believe this sensitivity to Gaussian noise is an inherent characteristic of diffusion models (considering their training methodology). The suggestion of robust optimization during the inversion process is a valuable direction for future work. However, such optimization might involve substantial computational complexity, and we currently lack concrete ideas for its implementation. A naive approach involves fine-tuning the model in an adversarial environment, though this would obviously compromise generation quality.
>
> Alternatively, we found that independent additive noise, including Gaussian noise, is readily distinguishable by a model, and applying traditional filtering methods before inversion is quite effective at removing it. To illustrate this, we trained a four-layer CNN for an experiment, yielding the following results:
>
> |  | Clean (TPR/Bit Acc) | GauNoise (TPR/Bit Acc) | S&PNoise (TPR/Bit Acc) | Other Noise (TPR/Bit Acc) | Adv ave. (TPR/Bit Acc) |
> | --- | --- | --- | --- | --- | --- |
> | Without Classifier&Filter | 1.000/1.000 | 0.988/0.8982 | 0.998/0.9487 | 0.999/0.9913 | 0.998/0.9762 |
> | With Classifier&Filter | 1.000/1.000 | 0.996/0.9495 | 1.000/0.9971 | 0.999/0.9909 | 0.999/0.9870 |
>
> These results show that this plug-and-play module improves performance under Gaussian and Salt-and-Pepper noise while having only a marginal effect on other distortions. In our experiment, the classifier successfully detected the target noise with a 94.9% probability, with a false positive rate of only 2.6%.
>
> ---
>
> ### **Re: Further estimation of the diversity (Question 4)**
>
> In the context of measuring diversity, we employ LPIPS in a non-standard manner. While LPIPS usually measures the direct visual distance between two images (where a smaller value means more similarity and often better visual quality), we calculate it differently for diversity. For $P$ prompts, and for each prompt generating $N$ images, we compute the visual distance for all $C_N^2$ possible pairs. We then average these distances for each prompt and finally average across all $P$ prompts, formally:
>
> $D = \frac{1}{P}\sum_{i=1}^{P} \frac{1}{C_N^2} \sum_{j,k,1\leq j<k \leq N} d(X_{i,j}, X_{i,k})$
>
> If the average distance between multiple images generated for the same prompt is small, we consider them similar and thus lacking diversity. Conversely, a larger average distance indicates better diversity. The function $d(·)$ can also be replaced by other distance metrics like CLIP-Image similarity, where a larger value indicates more similarity; therefore, a smaller value would indicate better diversity. Specific results are in the table below.
>
> We also followed your suggestion and conducted a user study. We randomly selected 20 prompts for each of 25 participants. For every prompt, we provided a set of five images generated by different watermarking methods and a non-watermarked baseline. Participants were asked to choose the set they perceived as most diverse. Our results, consistent with the quantitative diversity measurements in the paper, are shown below, and we hope this addresses your concerns.
>
> Our conclusions on diversity align with previous work. PRC, as a cryptographically undetectable watermark, exhibits diversity nearly comparable to unwatermarked images, while GS shows the lowest diversity due to its fixed key. Our undetectability experiments (Section 4.3) further support this, as the classifier easily distinguishes between GS and unwatermarked images by learning common features.
>
> |  | Diversity-LPIPS-SDP ↑ | Diversity-LPIPS-COCO ↑ | Diversity-CLIPI-SDP ↓ | Diversity-CLIPI-COCO ↓ | User-Study-WR ↑ |
> | --- | --- | --- | --- | --- | --- |
> | SD v2.1 | 0.6756 | 0.7072 | 0.7085 | 0.6949 | 0.174 |
> | StableSig | 0.6667 | 0.6917 | **0.7034** | 0.7051 | 0.140 |
> | TRW | 0.6655 | 0.6943 | 0.7152 | 0.6939 | 0.128 |
> | GS | 0.6156 | 0.6446 | 0.7364 | 0.7123 | 0.112 |
> | PRCW | **0.6747** | **0.7074** | ***0.7095*** | **0.6913** | **0.240** |
> | T2SMark | ***0.6746*** | ***0.7069*** | 0.7112 | ***0.6939*** | ***0.206*** |
>
> ---
>
> ### **Re: Can T2SMark preserve similarity to non-watermarked results? (Question 5)**
>
> Yes, T2SMark successfully preserves the similarity between watermarked and non-watermarked generated results. We performed t-tests on CLIP and FID scores, just like GS did, and our results indicate a very small difference between our watermarked images and their unwatermarked counterparts.
>
> In contrast, GS's diversity is actually quite limited. Our visual results in Appendix C show that GS-generated images consistently have similar layouts, often with a black shadow in the bottom-left area, a problem absent in our method. Additionally, the results in table above further confirm GS's relatively poor diversity. This lack of diversity can significantly affect user experience, especially when users generate multiple images from the same prompt and expect varied outputs to choose from. If the images share similar layouts, it considerably diminishes their selection options and overall satisfaction.
>
> ---
>
> ### **Re: Can the proposed method be extended to the Diffusion Transformer method? (Question 6)**
>
> Yes! We have shown our method's compatibility with the Diffusion Transformer architecture in our generalization experiments (Section 4.4). There, we compared our approach with other inversion-based methods on SD 3.5M, which is based on the Diffusion Transformer. Our results demonstrate that our method still achieves state-of-the-art performance on this architecture.

---

> > ### Comment · Reviewer_bX8p · 2025-08-07
> >
> > Can you provide a more detailed explanation of the two-stage encoding process? I'm confused in Figure 3 because it doesn't clearly indicate which components belong to stage one and stage two.
> >
> > Additionally, if the session key is not derived from the master key, then what is the relationship between them?
> > I think the session key is generated by the PRNG (which I believe stands for Pseudorandom Number Generator, though the paper doesn't define this abbreviation), and the master key can control the seed for PRNG.
> >
> > While the encoding box in the lower left of Figure 3 doesn't explicitly indicate whether the PRNG output serves as the session key?

---

> > > ### Author Response · Authors · 2025-08-07
> > >
> > > Thanks for the reply! We provide our responses to each point below.
> > >
> > > ---
> > >
> > > ### **Detailed Explanation of the Two-Stage Encoding Process**
> > >
> > > The encoding algorithm is the same for both stages and is quite complex, so we've shown it in detail in the bottom-left of Figure 3. The process takes two parameters: the **Message** (the content to be encoded, which is always at the start of the encoding arrow) and the **Key** (which always comes from below the arrow). The output is the encoded noise vector, which is always at the end of the arrow.
> > >
> > > - In **Stage One**, we use the master key as the **Key** and the session key as the **Message**, encoding them to produce the noise vector **$z_T^k$**.
> > > - In **Stage Two**, we use the session key as the **Key** and the watermark content as the **Message**, encoding them to produce the noise vector **$z_T^b$**.
> > >
> > > Therefore, the process involving **$z_T^k$** belongs to Stage One, and the process involving **$z_T^b$** belongs to Stage Two. This is consistent with our description in **Section 3.5**.
> > >
> > > In the specific encoding process, the **Key** acts as the seed for a PRNG to generate intermediate variables
> > > $\{\boldsymbol{v_i}\}_{i=1}^m$ and $\boldsymbol{w}$.
> > > The **Message** is then directly encoded with $\{\boldsymbol{v_i}\}\_{i=1}^m$ and  $\boldsymbol{w}$ via **Equation 4** to form the noise vector.
> > >
> > > ---
> > >
> > > ### **Relationship Between the Session Key and the Master Key**
> > >
> > > The generation of the session key is independent of the master key. A new session key is randomly generated for every image. In **Stage One**, we use the master key and the session key  to produce the noise vector **$z_T^k$**.
> > >
> > > ---
> > >
> > > ### **Whether the output of the PRNG serves as the session key?**
> > >
> > > No, the PRNG's output consists of the intermediate variables $\{\boldsymbol{v_i}\}_{i=1}^m$ and  $\boldsymbol{w}$, which are directly used in the encoding and decoding processes (**Equations 4 and 5**). The session key acts as a parameter in both stages—it is the content being encoding in Stage One and the key for encoding in Stage Two. Notably, the session key is not generated during the encoding process itself, but randomly generated for each image.
> > >
> > > ---
> > >
> > > To make our method clearer in the paper:
> > >
> > > - We will define the abbreviation PRNG in the text.
> > > - We will update Figure 3 to explicitly indicate that the session key is derived from an independent entropy source.
> > > - We will update Figure 3 to explicitly show how the PRNGs are used and improve the readability of the related descriptions.

---

> > > > ### Comment · Reviewer_bX8p · 2025-08-08
> > > >
> > > > Thanks for the reply.
> > > >
> > > > I think the explanation in Figure 3 can be further clarified. It would be helpful to explicitly show that the key in the encoding process is the master key in Stage 1 and the session key in Stage 2. Similarly, the message in Stage 1 is the session key and in Stage 2, it is the watermark. Otherwise, the current figure may cause some confusion regarding the roles of the key and message in the encoding process.
> > > >
> > > > Additionally, while LPIPS is a commonly used metric for perceptual similarity, I still believe it serves as an indirect measure of diversity. A more intuitive and direct measure of diversity would be to compute the variance of the generated watermarked images.
> > > > Can you consider including this type of variance-based analysis for the watermarked images in your evaluation? For example, variance measurement in the feature space.

---

> > > > > ### Author Response · Authors · 2025-08-08
> > > > >
> > > > > Thank you for reply!
> > > > >
> > > > > We will revise Figure 3 according to your suggestion to clarify the usage of the master key and the session key.
> > > > >
> > > > > Regarding the diversity evaluation, we have performed the measurements you requested on the images we previously generated using the COCO dataset. The results are as follows:
> > > > >
> > > > > |  | SD v2.1 | StableSig | TRW | GS | PRCW | T2SMark |
> > > > > | --- | --- | --- | --- | --- | --- | --- |
> > > > > | Variance of CLIP Embeddings (×$10^5$) | 59.60 | 57.59 | 59.16 | 56.19 | 59.86 | 59.77 |
> > > > >
> > > > > The results are highly consistent with those presented in our main paper and rebuttal. T2SMark and PRCW maintain a level of diversity comparable to the watermark-free model, while GS performs the worst. We have already substantiated our conclusions through multiple methods, including LPIPS, CLIP, and user studies.
> > > > >
> > > > > We are happy to address any further questions you may have. We hope they can be raised as soon as possible, as the discussion period is nearing its end and we need sufficient time to conduct any related experiments.

---

> > > > > > ### Comment · Reviewer_bX8p · 2025-08-09
> > > > > >
> > > > > > Thanks for the reply. This validation of CLIP embedding variance is sufficient enough to address my concern for diversity evaluation. I will raise my score rating. And I also suggest including more visualization samples in the final version to enhance the presentation for diversity, since it's a key concept of this work.

---

> > > > > > > ### Author Response · Authors · 2025-08-09
> > > > > > >
> > > > > > > Dear Reviewer bX8p, thank you for your positive feedback! We are glad our reply addressed your concern regarding diversity. We will also add more visualization samples to the final version as you suggested. Thanks again for helping us improve the paper!

---

### Official Review · Reviewer_eVQe · 2025-07-03

**Clarity:** 3
**Significance:** 3
**Originality:** 3
**Rating:** 4
**Confidence:** 4

**Summary:**

Problem:
A challenge for existing NaW methods, such as Gaussian Shading (GS) and PRC-Watermark (PRCW), lies in obtaining an optimal balance between watermark robustness and generation diversity. Some methods (ex: GS) heavily constrain initial noise sampling and therefore prioritize strong robustness which limits the variability of generated images. Other methods (ex: PRCW) that preserve diversity are sometimes fragile for real-world deployment as they fail under common distortions.

Method
- To address the above issue, the paper introduces T2SMark, a novel two-stage watermarking scheme which is based on Tail-Truncated Sampling (TTS).
- T2SMark's design has two primary innovations:
   1. TTS enhances robustness by embedding bits exclusively within the reliable tail regions of the Gaussian distribution, while randomly sampling the central zone to maintain the latent distribution. This approach is different from previous methods that just map bits to positive or negative values - using the statistical properties of the noise for improved resilience.
   2. The two-stage framework ensures sampling diversity by integrating a randomly generated session key into both encryption pipelines - Static master key encrypts this random session key in the first stage, which then encrypts the actual watermark bits in the second stage. This encryption introduces controlled randomness across the full noise vector and preserves generation diversity.

Evaluation
- The method is evaluated on diffusion models with both U-Net and DiT backbones which demonstrates its ability to achieve an optimal balance between robustness and diversity.
- The method shows better traceability and competitive detection performance, while effectively preserving image quality and generalizability across different diffusion model architectures.

**Questions:**

1. "T2SMark is highly vulnerable to Gaussian noise, which is a common real world perturbation that challenges its optimal balance for real world deployment" - Please discuss any strategies to mitigate this.
2. "Bit accuracy drops rapidly with longer session keys due to cascading errors, and the 16 bit key's sufficient entropy" - this claim needs stronger justification for adversarial environments. Provide a more rigorous security analysis of the session key length and explain how imperfect session key recovery impacts traceability and detection.
3. Visual Results - please provide more example images in Appendix C visual results (preferably images with text in them) to clearly show diversity and that image quality is maintained. It would also be helpful to add SSIM and PSNR metrics for image quality.

**Ethical Concerns:**

["NO or VERY MINOR ethics concerns only"]

**Final Justification:**

The authors have addressed all my questions. I will maintain my rating.

**Limitations:**

Yes

**Quality:**

3

**Strengths And Weaknesses:**

STRENGTHS:
1. Principled method:
    - TTS is introduced to enhance robustness by embedding bits in the more reliable tail regions of the Gaussian distribution - this is supported by theoretical BEP analysis in Appendix A which shows that truncation decreases bit error probability.
    - The two stage key hierarchy ensures generation diversity and robustness by using a random session key, encrypted by a master key, to then encrypt the watermark bits, injecting randomness across the noise vector. For decoding, multidimensional projections of the reconstructed Gaussian noise are used to recover watermark bits robustly.
2. Comprehensive experiments
    - Comparison against multiple baselines - traditional post processing, fine tuning, and SOTA inversion based techniques.
    - Multiple metrics (TPR, Bit Acc, LPIPS, CLIP score, FID) are used for evaluation, and undetectability is assessed via a ResNet classifier.
    - Robustness is evaluated on 9 distortion types
    - Evaluation on both SD 2.1 (UNet) and SD 3.5 M (DiT) demonstrates the method's generalizability across different diffusion model architectures.
3. The results clearly show T2SMark's balance across robustness, diversity, and quality, many times outperforming or matching baselines.
4. Sufficient implementation details are provided for reproducibility - specific SD versions, guidance scales, denoising steps, PyTorch version, GPU used and relevant Github codes.

WEAKNESSES
1. T2SMark is highly vulnerable to Gaussian noise, even at low intensities which is a practical concern for real-world deployment and a common limitation for inversion-based methods.
2. The ablation study shows rapid bit accuracy drop for longer session keys due to cascading errors, raising scalability questions for high-entropy keys.
3. Empirical tuning of parameters like truncation threshold τ suggests potential sensitivity to new model architectures, requiring re-tuning for optimal performance.

---

> ### Author Rebuttal · Authors · 2025-07-31
>
> Thank you for your time and valuable feedback on our submission. We have carefully considered each point raised and address them individually below.
>
> ---
>
> ### **Re: Vulnerability to Gaussian noise and corresponding solutions (Weakness 1 & Question 1)**
>
> We acknowledge T2SMark's vulnerability to Gaussian noise. While this is a limitation, it is important to note that our method's performance gap compared to baseline methods under Gaussian noise is marginal (only 0.53% lower than GS, shown in Table 10).
>
> We attribute this sensitivity primarily to the inherent nature of diffusion models (considering their training process), and our two-stage watermarking framework's cascading errors could amplify this effect. Given the diversity gains from our two-stage approach, we consider this a justifiable trade-off.
>
> As for mitigation, any general method for improving robustness (e.g., adding error-correcting codes) can alleviate this. To specifically address the issue of additive noise like Gaussian and Salt-and-Pepper, we found that such noise is readily distinguishable by a simple model. Furthermore, applying traditional filtering methods *before* inversion proves highly effective at removing this noise.
>
> To demonstrate this, we trained a four-layer convolutional neural network for noise detection. If the target noise is detected, the image is first filtered and then inverted. The results are as follows:
>
> |  | Clean (TPR/Bit Acc) | GauNoise (TPR/Bit Acc) | S&PNoise (TPR/Bit Acc) | Other Noise (TPR/Bit Acc) | Adv ave. (TPR/Bit Acc) |
> | --- | --- | --- | --- | --- | --- |
> | Without Classifier&Filter | 1.000/1.000 | 0.988/0.8982 | 0.998/0.9487 | 0.999/0.9913 | 0.998/0.9762 |
> | With Classifier&Filter | 1.000/1.000 | 0.996/0.9495 | 1.000/0.9971 | 0.999/0.9909 | 0.999/0.9870 |
>
> These results indicate that this plug-and-play module improves performance under Gaussian and Salt-and-Pepper noise with only a minimal impact on other distortions. In our experiments, the classifier successfully detected the target noise with a 94.9% probability while maintaining a low false positive rate of only 2.6%.
>
> ---
>
> ### **Re: Scalability of the session key (Weakness 2 & Question 2)**
>
> Regarding the statement "a 16-bit key provides sufficient entropy," our original intent was to refer to its sufficiency for generating diversity. For individual users, ensuring distinct session keys allows for the generation of up to $2^{16}=65,536$ images without repeating a codeword, thereby guaranteeing diversity. While enterprise users might require a larger keyspace, our method still maintains competitive performance with 24-bit and even 32-bit keys.
>
> For traceability, the session key serves as the crucial seed for a pseudo-random number generator used in the second-stage decryption. Even a minor alteration to this seed will cause the pseudo-random sequence to diverge significantly, rendering the second-stage decoded watermark meaningless. Therefore, perfect recovery of the session key is paramount for successful traceability.
>
> Regarding detection, we designed this functionality to operate in the first stage, which inherently protects it from the impact of cascading errors. As shown in Table 7, detection performance exhibits very minimal variation with changes in session key length.
>
> ---
>
> ### **Re: More visual results (Question 3)**
>
> Current NeurIPS 2025 policy does not allow us to provide additional visual evidence directly within the rebuttal or discussion phase. However, we are fully committed to including more example images, particularly those with embedded text, in Appendix C if the paper is accepted, to clearly showcase both diversity and maintained image quality.
>
> As compensation and to address your concern regarding diversity, we conducted a user study. We randomly selected 20 prompts for each of 25 participants. For every prompt, we presented a set of five images generated by different watermarking methods and also a non-watermarked baseline. Participants were then asked to choose the set they perceived as most diverse. Our results, presented below, largely align with our quantitative diversity measurements in the paper, which we hope alleviates your concerns:
>
> | Method | SD v2.1 | StableSig | TRW | GS | PRCW | T2SMark |
> | --- | --- | --- | --- | --- | --- | --- |
> | Win Rate | 0.174 | 0.140 | 0.128 | 0.112 | 0.240 | 0.206 |
>
> Regarding SSIM and PSNR, these are *relative* metrics that measure differences from an original image. As a NaW method, T2SMark encodes watermarks directly into the initial noise; thus, there is not a strict "original image" counterpart to compare against. Instead, image quality is assessed by evaluating the overall quality of the generated content. In this paper, we generated a total of 10 sets comprising 10,000 images. Similar to GS, we performed t-tests on CLIP and FID scores against unwatermarked images for all methods. A smaller t-value indicates greater indistinguishability from unwatermarked images. The t-test results are provided in Table 1 and Table 3, with further details in Appendix B.3. Appendix B.5 also offers results on additional datasets. We believe this provides sufficient evidence of maintained image quality.
>
> ---
>
> ### **Re: Empirical tuning of parameters (Weakness 3)**
>
> For practical deployment, our hyperparameters offer a broad selection range. Our experiments  show the truncation threshold (τ) performs poorly only at extreme values (too large, reducing redundancy; too small, leading to bit flips and degrading to normal sampling). It maintains similar performance across most of its range (Table 8), and we did not pick the absolute optimal threshold. Similarly, the chosen session key channel has limited impact (Table 9).
>
> Our generalization experiments (Section 4.4) further support this: we achieved SOTA performance (Table 3) on new architectures without any empirical hyperparameter re-tuning. More detailed experimental settings about it can be found in Appendix B.1.

---

> > ### Comment · Reviewer_eVQe · 2025-08-06
> >
> > The authors have addressed my questions. I will maintain my score.

---

> ### Author Response · Authors · 2025-08-06
> **Thanks for your comment!**
>
> Dear Reviewer eVQe, we are pleased to have addressed your questions. Thank you for your positive evaluation of our work!

---

### Comment · Area_Chair_oKUQ · 2025-08-04
**Please read the rebuttal and start the discussion**

Dear Reviewers and Authors,

Thank you all for your efforts so far. As the author–reviewer discussion period will conclude on **August 6**, please start the discussion as soon as possible.


**For Reviewers:**
Please read the authors’ responses and, if necessary, continue the discussion with them.

* If your concerns have been addressed, consider updating your review and score accordingly.

* If some concerns remain, or if you share concerns raised by other reviewers, clearly state these in your review and consider adjusting your review (positively or negatively).

* If you feel that your concerns have not been addressed, you may also choose to keep your review as is.

* I will follow up with you again during the reviewer–AC discussion period (August 7–13) to finalize the reviews and scores.


**For Authors:**
If you have not already done so, please respond to all questions raised by the reviewers. Keep your responses factual, concise, and ensure that every point raised is addressed.

Best regards,

The AC

---

### Note · Authors · 2025-08-11

We sincerely appreciate the efforts of the AC and all reviewers. We are grateful to the reviewers for their recognition of our work’s direction and performance, and we are pleased that we were able to address most of their concerns during the rebuttal and discussion phases, particularly the following three shared points:

- Vulnerability to Gaussian Noise: This is a common issue for inversion-based watermarks, and we acknowledge it in the paper. Our results show our method's bit accuracy is only 0.53% below the best-performing method  under this distortion. We believe this is due to the denoising network's inherent sensitivity to Gaussian noise (considering its training). With supporting experimental results, we actively discussed effective mitigation strategies, including error-correcting codes and a detection-and-filtering pre-processing approach.
- Robustness of Hyperparameter Selection: The hyperparameters in our paper were chosen empirically, which raised concerns about their transferability to different models. Based on the experimental results in our appendix, we explained that the range of acceptable hyperparameters is flexible. This point was further substantiated by our generalization experiments on SD v3.5M.
- Choice of Diversity Metrics: Reviewers were concerned about using LPIPS to measure diversity. We clarified that we use it in an unconventional way, as a measure of the average distance between a set of generated images. We also conducted additional experiments with CLIP Image similarity and a user study, and all results aligned with our LPIPS findings, further validating our diversity claims.

We also thank the reviewers for their constructive suggestions. In the final manuscript, we will:

- Revise Figure 1 to use quantitative data instead of an abstract representation.
- Update Figure 3 to clearly illustrate the roles of the master and session keys.
- Provide more visual examples.
- Include an analysis of computational overhead.
- Add a discussion of robustness against geometric distortions.
- Integrate other key elements discussed during the rebuttal.

Thanks again to the AC and reviewers for helping us make this work better!

---

### Decision · Program_Chairs · 2025-09-17

**Decision:**

Accept (poster)

**Comment:**

The paper proposes T2SMark, a watermarking framework for diffusion models that achieves a balance between robustness to distortions and generation diversity. The method employs a two-stage approach: a tailored inversion process for watermark embedding and a tail-truncated sampling (TTS) strategy to improve robustness.

All reviewers were actively engaged in the rebuttal and agreed that this work represents a solid contribution. Most of the concerns raised during the review process have been satisfactorily addressed. The authors are encouraged to incorporate the reviewers’ suggestions and feedback, as summarized in the final remarks, into the camera-ready version.